# DevFD: Developmental Face Forgery Detection by Learning Shared and Orthogonal LoRA Subspaces

**Tianshuo Zhang**[1,2]    **Li Gao**[3]    **Siran Peng** [1,2]    **Xiangyu Zhu**[1,2*]   **Zhen Lei**[1,2,4,5*]

[1] School of Artificial Intelligence, University of Chinese Academy of Sciences, Beijing, China
[2] MAIS, Institute of Automation, Chinese Academy of Sciences, Beijing, China
[3] China Mobile Financial Technology Co., Ltd., Beijing, China
[4] CAIR, HKSIS, Chinese Academy of Sciences, Hong Kong, China
[5] School of Computer Science and Engineering, the Faculty of Innovation Engineering,
M.U.S.T, Macau, China
`tianshuo.zhang@nlpr.ia.ac.cn, gaolids@chinamobile.com`
`pengsiran2023@ia.ac.cn, xiangyu.zhu@ia.ac.cn, zhen.lei@ia.ac.cn`

## Abstract

The rise of realistic digital face generation and manipulation poses significant social risks. The primary challenge lies in the rapid and diverse evolution of generation techniques, which often outstrip the detection capabilities of existing models. To defend against the ever-evolving new types of forgery, we need to enable our model to quickly adapt to new domains with limited computation and data while avoiding forgetting previously learned forgery types. In this work, we posit that genuine facial samples are abundant and relatively stable in acquisition methods, while forgery faces continuously evolve with the iteration of manipulation techniques. Given the practical infeasibility of exhaustively collecting all forgery variants, we frame face forgery detection as a continual learning problem and allow the model to develop as new forgery types emerge. Specifically, we employ a Developmental Mixture of Experts (MoE) architecture that uses LoRA models as its individual experts. These experts are organized into two groups: a Real-LoRA to learn and refine knowledge of real faces, and multiple Fake-LoRAs to capture incremental information from different forgery types. To prevent catastrophic forgetting, we ensure that the learning direction of Fake-LoRAs is orthogonal to the established subspace. Moreover, we integrate orthogonal gradients into the orthogonal loss of Fake-LoRAs, preventing gradient interference throughout the training process of each task. Experimental results under both the datasets and manipulation types incremental protocols demonstrate the effectiveness of our method.

## 1   Introduction

The swift evolution of generative models, including Generative Adversarial Networks (GANs) [2, 38, 62] and Diffusion models [17, 18, 61], along with the rise of large models [4, 25, 50, 63, 71], propels the rapid advancement of face forgery technology, which introduces significant security risks and contributes to a crisis of public trust. These manipulation methods leave unique traces of forgery. Existing detectors perform well with known forgery types but struggle to identify novel ones. When new manipulation types arise, a trivial method is to add new forgery data to the primary dataset and retrain a binary model. However, the rapid evolution of forgery techniques necessitates a new learning strategy that can quickly adapt to new forgeries using limited computation and data.

---

[*]Corresponding Authors.

39th Conference on Neural Information Processing Systems (NeurIPS 2025).

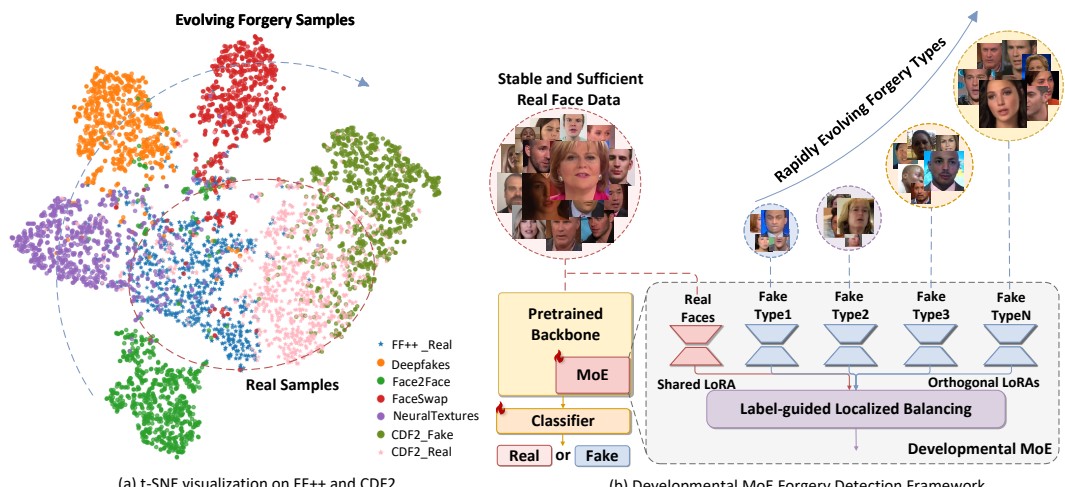

(a) t-SNE visualization on FF++ and CDF2.   (b) Developmental MoE Forgery Detection Framework

Figure 1: (a) The t-SNE visualization of features extracted from the baseline on FF++ [48] and CDF2 [27] shows that real faces exhibit a close distribution, while fake faces form five distinct clusters. This observation inspires us to (b) propose a developmental mixture of experts to model the continuous emergence of unknown fake faces using a set of orthogonal LoRA subspaces, while concurrently employing a dedicated LoRA to preserve the commonalities of authentic faces. A label-guided localized balancing strategy employs the LoRA sequence to separately model the common real faces and the incremental fake types information.

In contrast to the rapidly evolving fake faces, real faces possess distinct characteristics. Genuine facial data is relatively abundant, stable, and acquired through a single modality, such as camera imaging. In the context of forgery detection, it typically does not introduce bias. These properties, which remain invariant as forgery techniques evolve, endow real faces with commonalities that may easily be overlooked. Fig. 1(a) presents a simple t-SNE [57] visualization experiment, using the baseline [8] model to extract features from both the FF++ [48] and CDF2 [27] datasets. The results reveal two key observations: (1) Real face samples from both datasets exhibit close distributions, while (2) forged samples form five distinct clusters. These observations inspire us to enable the model to adapt to emerging manipulation methods while learning the commonalities of real faces.

In this paper, we approach face forgery detection as a continual learning problem and make the model "developmental," enabling it to adapt to new forgery types continuously. As shown in Fig. 1(b), we propose a developmental Mixture of Experts (MoE) architecture, DevFD, which employs Low-Rank Adaptation (LoRA) models as individual experts to fine-tune pre-trained models within different subspaces. As illustrated in Fig. 2, the model expands a new LoRA branch when adapting to an unknown forgery type. However, adapting to new forgery types can disrupt the knowledge in established subspaces, leading to catastrophic forgetting. To address this, we use an orthogonal loss to constrain the learning direction of new LoRAs to be orthogonal to all previously established LoRAs, thereby preserving the learned knowledge. This part of the LoRAs ultimately forms an orthogonal sequence termed Fake-LoRAs. For real faces that are stable and exhibit common properties, modeling them with the same sequence overlooks their overall distribution. Therefore, we introduce a separate shared expert, termed Real-LoRA, to fine-tune and refine knowledge about real faces. To enable the Real-LoRA and Fake-LoRAs to align with their assigned roles and avoid disrupting the orthogonality of Fake-LoRAs, we design a label-guided localized balancing strategy that softly constrains the experts through a weighted response matrix. For unknown forgery images, the router aggregates the outputs of all LoRAs to make a joint decision, thereby maintaining collaboration among the experts.

Additionally, we theoretically analyze that the subspace orthogonal loss still causes forgetting due to the early stages of optimization not satisfying the orthogonality condition, where learning new tasks disrupts the established knowledge of other subspaces. To avoid this, we integrate orthogonal gradients into the orthogonal loss and propose a new integrated orthogonal loss to constrain the Fake-LoRAs. Experiments demonstrate that our method achieves state-of-the-art average accuracy and the lowest average forgetting rate in continual learning experiments on both the datasets incremental protocol and the manipulation types incremental protocol.

**Contributions.** We summarize our contributions as follows:

- We propose a developmental MoE architecture to address the continuous emergence of new forgery types. We use a label-guided localized balancing strategy to allocate Real-LoRA to model the real face information, while Fake-LoRAs capture the incremental fake face information.
- Based on theoretical analysis, we integrate orthogonal gradients into the subspace orthogonal loss and form a new integrated orthogonal loss to mitigate the interference of gradients on the knowledge of established subspaces during the entire stages of training.
- Extensive experiments demonstrate that our proposed method achieves the highest average scores and the lowest average forgetting rate among all comparison methods.

## 2 Related Works

### 2.1 Face Forgery Detection

Face forgery detection requires the model to classify input images or videos as either real or fake. Related methods [5, 6, 10, 33, 58, 67] focus on identifying forgery cues present in the forged media and can be categorized into two major groups. The first category designs models that incorporate additional information to aid in detection, such as frequency analysis [13, 24, 35, 42], wavelet [36], graphics [72, 73], language descriptions [54], and audio-video consistency [39]. The second category [23, 26, 51] is data-driven, utilizing only real faces and training models through a self-supervised approach. With the continuous evolution of forgery methods, continual learning is applied to forgery detection. DFIL [40] and HDP [55] treat the learning of forgery datasets as a sequence of sub-tasks, requiring the model to learn sequentially across these tasks, achieving satisfactory performance. In this work, we design a Developmental MoE to address the challenges of continual learning in face forgery detection.

### 2.2 Continual Learning

Continual learning typically models the task as a sequence of sub-tasks, requiring the model to adapt to dynamic data distributions incrementally. The goal is to achieve high performance on newly learned tasks while avoiding catastrophic forgetting of information from previous tasks. Related approaches can be broadly divided into three categories: Regularization-based Approaches [47, 49, 53, 59, 65, 70], Replay-based Approaches [22, 34, 45, 46, 52], and Optimization-based Approaches [11, 21, 31, 32, 41]. Among the latest methods, OrCo [1] enhances generalization in class-incremental learning through orthogonality and contrast. Meanwhile, MoECL [68] adapts to new tasks by introducing an MoE. DFIL [40] and SUR-LID [7] achieve satisfactory results using replay mechanisms. In this work, we treat the iteration of fake faces as an incremental learning problem and employ a developmental architecture to facilitate continual learning.

### 2.3 Mixture of Experts

The Mixture of Experts (MoE) is a set of sparse models with multiple experts and a routing network. By training MoE in parallel, methods [14, 37, 74] enable efficient fine-tuning of large models with minimal computational cost. MoECL [68] introduces MoE to continual learning, employing an intuitive trainable-freezing strategy to reduce the model's forgetting rate. Low-rank Adaptation (LoRA) [15] further decreases parameter and computational cost by mapping inputs into a low-rank subspace. O-LoRA [60] utilizes LoRAs as experts and constrains the orthogonality of LoRA subspaces. InfLoRA [30] further applies orthogonal LoRA to vision tasks, improving the performance of multiple sub-tasks through pre-defined orthogonal subspaces. MoEFFD [20] integrates MoE and LoRA into face forgery detection, allowing each expert to specialize in extracting features related to different forgery techniques. We use the LoRA module to construct an MoE architecture to address continual face forgery detection.

## 3 Methodology

We cast forgery detection as a continual learning problem $\mathcal{D} = \{\mathcal{D}_1, \ldots, \mathcal{D}_T\}$, where $T$ is the total number of tasks. For any task $\mathcal{D}_t$, let $\mathbf{x}_{i,t}$ be the sample and $y_{i,t}$ its corresponding label, and $n_t$ represent the number of samples in the current task. The model sequentially learns each task

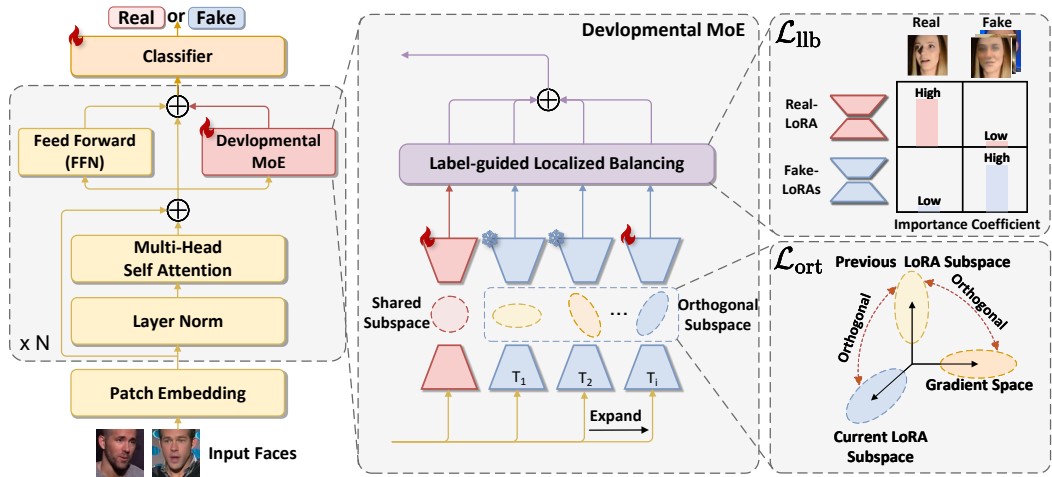

Figure 2: The proposed DevFD framework employs a Developmental MoE architecture to fine-tune the FFN layer in each transformer block. The architecture establishes a developmental LoRA sequence, which adds new branches as the number of tasks increases, enabling the model to handle emerging new types of forgeries. A new label-guided localized balancing strategy allocates the LoRA sequence into two purposes: the Real-LoRA fine-tune and refine knowledge about real faces, while the Fake-LoRAs compose an orthogonal sequence to model the unique cues of fake faces. We integrate orthogonal gradients into the orthogonal loss to alleviate the interference of gradients on previously learned tasks during the training phase when orthogonality is not yet achieved, thereby achieving a lower rate of forgetting.

$\mathcal{D}_t = \{(\mathbf{x}_{i,t}, y_{i,t})\}_{i=1}^{n_t} = \{\mathcal{X}_t, \mathcal{Y}_t\}$ in the task sequence $\mathcal{D}$ to adapt to new types of forgery. Based on the characteristics of the face forgery detection task, we provide the following descriptions:

$$\begin{cases} p(\mathcal{X}_i) \neq p(\mathcal{X}_j), \ \mathcal{Y}_i = \mathcal{Y}_j = \text{Fake for } i \neq j, \\ p(\mathcal{X}_i) = p(\mathcal{X}_j), \ \mathcal{Y}_i = \mathcal{Y}_j = \text{Real for } i \neq j, \end{cases} \tag{1}$$

where $p(\cdot)$ is the sample distribution, $\mathcal{X}$ is the samples and $\mathcal{Y}$ is the labels. For fake faces, different forgery types belong to different domains but share the same label $\mathcal{Y} = $ "Fake", presenting a domain incremental problem. In contrast, due to the uniform acquisition methods and the stable and abundant samples of real faces, real faces across datasets exhibit similarity. We aim to model and align them into a unified distribution, i.e., $p(\mathcal{X}_i) = p(\mathcal{X}_j)$. We strictly isolate training data across tasks.

## 3.1 Developmental MoE Architecture

We propose a novel developmental MoE architecture using LoRAs as individual experts to fine-tune pre-trained models within different subspaces; the overview is shown in Fig. 2. First, we employ a Vision Transformer (ViT) pre-trained on real faces as the backbone, denoted as $g_{\mathbf{\Phi}}(f_{\mathbf{\Theta}}(\cdot))$, where $f_{\mathbf{\Theta}}$ represents the pre-trained ViT network, and $g_{\mathbf{\Phi}}$ denotes the classifier. We freeze the parameter $\mathbf{\Theta}$ of the ViT backbone $f$ and introduce LoRA to fine-tune the Feed Forward Network (FFN) weights in each transformer block. For a parameter matrix of any FFN layer $\mathbf{W} \in \mathbb{R}^{d_O \times d_I}$ with input dimension $d_I$ and output dimension $d_O$, LoRA decomposes the parameter change matrix $\Delta W$ into the multiplication of two low-rank matrices:

$$\Delta \mathbf{W} = \mathbf{AB}, \tag{2}$$

where $\mathbf{B} \in \mathbb{R}^{d_I \times r}$ is the dimensionality reduction matrix and $\mathbf{A} \in \mathbb{R}^{r \times d_O}$ is the dimensionality expansion matrix. Generally, the ranks $r$ of the two matrices are equal and satisfy the condition: $r \ll \min(d_O, d_I)$. To enable the model to adapt to each task, as illustrated in Fig. 2, we design a developmental LoRA sequence as MoE adapters:

$$\begin{aligned} \mathbf{e} &= \text{FFN}(\mathbf{x}) + \text{MoE}(\mathbf{x}) \\ &= \text{FFN}(\mathbf{x}) + \sum_{j=0}^{t} \text{LLB}(\mathbf{A}_j \mathbf{B}_j \mathbf{x}). \end{aligned} \tag{3}$$

Here, $\mathbf{x}$ is the input, while $\mathbf{e}$ is the output. The LoRA modules in the sequence $\{\mathbf{A}_j\mathbf{B}_j\}_{j=0}^{t}$ are assigned by a label-guided localized balancing strategy (LLB) for two purposes: the Real-LoRA $\mathbf{A}_0\mathbf{B}_0$ is designed to learn and refine the common information of real faces modeled by the backbone, while the Fake-LoRAs $\{\mathbf{A}_j\mathbf{B}_j\}_{j=1}^{t}$ compose an orthogonal sequence for learning fake face information specific to each task.

## 3.2 Learning LoRA Subspaces enhanced by Orthogonal Gradients

The reason for catastrophic forgetting is that parameter adjustments while learning the current task disrupt the knowledge acquired from previous tasks [30]. The subspace established by LoRA can approximately represent the information acquired for a new task [60]. As a developmental structure, DevFD constrains the learning direction of the expanded branches for the new task to prevent interference with previously established subspaces by using an integrated orthogonal loss enhanced with orthogonal gradients. First, the learning of the t-th task can be modeled as follows:

$$\mathbf{e} = \mathbf{W}\mathbf{x} + \sum_{j=1}^{t} \mathbf{A}_j\mathbf{B}_j\mathbf{x} = \mathbf{W}_{t-1}\mathbf{x} + \mathbf{A}_t\mathbf{B}_t\mathbf{x} = \mathbf{W}_t\mathbf{x}, \tag{4}$$

where $\mathbf{W}$ is the parameter matrix of the FFN layer, $\mathbf{e}$ and $\mathbf{x}$ represent the output and input of the network, respectively. Existing methods attempt to use orthogonal loss to control the learning direction of LoRA to be orthogonal to the established subspaces in order to prevent forgetting. However, the forgetting remains severe. We demonstrate that relying solely on the subspace orthogonal loss is insufficient and present an orthogonal gradient solution to further mitigate forgetting.

**Why Orthogonal LoRA Subspaces are Still Forgetful?** The adaptation of LoRA to new tasks involves fine-tuning the weight matrices within their respective subspaces. LoRA maps the input to the subspace : span $\{\mathbf{b}_1^t, \ldots, \mathbf{b}_r^t\}$, where the row vectors $\{\mathbf{b}_1^t, \ldots, \mathbf{b}_r^t\}$ of the dimensionality reduction matrix $\mathbf{B}_t$ serve as a basis for this subspace. O-LoRA [60] enforces the orthogonality between the basis $\{\mathbf{b}_1^t, \ldots, \mathbf{b}_r^t\}$ and the basis $\{\mathbf{b}_1^i, \ldots, \mathbf{b}_r^i\}$ of the forward LoRAs' subspace, ensuring that the subspaces of the LoRA sequence are mutually orthogonal. Let $\mathbf{O}_{\mathbf{i,t}}$ be the dot product of two bases of these subspaces. The orthogonality optimization objective can then be expressed as:

$$\underset{\{\mathbf{b}_1^t,\ldots,\mathbf{b}_r^t\}}{\arg\min} \|\mathbf{O}_{i,t}\|^2 = \underset{\{\mathbf{b}_1^t,\ldots,\mathbf{b}_r^t\}}{\arg\min} \left\|\mathbf{B}_t^T\mathbf{B}_i\right\|^2, \; i = 1, ..., t-1. \tag{5}$$

Notably, the orthogonality condition exists in the form of an optimization objective. We will demonstrate that preventing the disruption of the information in the previous subspaces depends on the strict $\mathbf{B}_t^T\mathbf{B}_i = 0$ throughout the training process. We begin with a proven proposition [30]: when the model learns the t-th task, fine-tuning $\mathbf{A}_t$ is equivalent to fine-tuning the original parameter matrix $\mathbf{W}$ within the subspace span $\{\mathbf{b}_1^t, \ldots, \mathbf{b}_r^t\}$, denoted as:

$$\Delta_{\mathbf{A}_t}\mathbf{W}_t = \Delta_{\mathbf{W}}\mathbf{W}_t\mathbf{B}_t^T\mathbf{B}_t, \tag{6}$$

where $\Delta_{\mathbf{A}_t}\mathbf{W}_t$ is the increment of the composed matrix $\mathbf{W}_t$ caused by the change of $\mathbf{A}_t$, and $\Delta_{\mathbf{W}}\mathbf{W}_t$ is the increment of the composed matrix $\mathbf{W}_t$ caused by the change of the original parameter $\mathbf{W}$. Thus, a projection matrix $\mathbf{B}_t^T\mathbf{B}_t$ projects each row vector of $\Delta_{\mathbf{W}}\mathbf{W}_t$ into the subspace span $\{\mathbf{b}_1^t, \ldots, \mathbf{b}_r^t\}$, it follows that the increment $\Delta_{\mathbf{A}_t}\mathbf{W}_t$ due to the fine-tuning of $\mathbf{A}_t$ is equivalent to mapping $\Delta_{\mathbf{W}}\mathbf{W}_t$ into span $\{\mathbf{b}_1^t, \ldots, \mathbf{b}_r^t\}$. In summary, the increment $\Delta_{\mathbf{A}_t}\mathbf{W}_t$ will not affect the subspace established by previous LoRA, under the assumption that span $\{\mathbf{b}_1^t, \ldots, \mathbf{b}_r^t\}$ is orthogonal to these subspaces, meaning that the orthogonality condition $\mathbf{B}_t^T\mathbf{B}_i = 0$ needs to hold strictly during the entire training process. Therefore, we argue that catastrophic forgetting occurs in the early stages of training on new tasks when the bases $\mathbf{B}_t$ and $\mathbf{B}_i$ are not perfectly orthogonal. The gradients of the new tasks can disrupt the established subspaces.

**Integrate Orthogonal Gradients into Orthogonal Loss.** To address the issues discussed above, we choose to directly constrain the gradient space while optimizing the orthogonality of subspaces, resulting in a new integrated orthogonal loss. First, it is proven that the gradient update of a linear or convolution layer lies within the span of the input vectors [29]. We place the proofs in the Appendix B. This proposition allows us to transform the constraints on the gradient space into one on the input space. Let the input matrix be $\mathbf{H} \in \mathbb{R}^{d_I \times n}$, where $n$ is the batch size. Due to the inconsistency in matrix dimensions, we first perform Singular Value Decomposition (SVD) on $\mathbf{H}_t^T$:

$$\mathbf{H}_t^T = \mathbf{U}_t\mathbf{\Sigma}_t\mathbf{V}_t^T. \tag{7}$$

We then choose the rows of $\mathbf{V}_t^T$ corresponding to the top-r singular values and compose $(\mathbf{V}_t^T)_r \in \mathbb{R}^{r \times d_I}$. We use $(\mathbf{V}_t^T)_r$ as an estimate of the gradient space and constrain it to be orthogonal to the previously established subspace. Let $\mathbf{G}_{\mathbf{i},\mathbf{t}}$ represent the dot product between the estimated gradient space and the previous subspace basis. The gradient constraint can then be expressed as:

$$\underset{\{\mathbf{b}_1^t,...,\mathbf{b}_r^t\}}{\arg\min} \|\mathbf{G}_{\mathbf{i},\mathbf{t}}\|^2 = \underset{\{\mathbf{b}_1^t,...,\mathbf{b}_r^t\}}{\arg\min} \|(\mathbf{V}_t)_r\mathbf{B}_i\|^2, \ i = 1,...,t-1. \tag{8}$$

In summary, we integrate the orthogonality of LoRA's subspace and the gradient space, designing the integrated orthogonal loss as follows:

$$\mathcal{L}_{\text{ort}} = \frac{1}{t-1} \sum_{i=1}^{t-1} \left( \lambda_1 \sum \|\mathbf{O}_{i,t}\|^2 + \lambda_2 \sum \|\mathbf{G}_{i,t}\|^2 \right). \tag{9}$$

Here, $\sum \|\cdot\|^2$ represents the sum of the squares of the matrix elements, $\lambda_1$ and $\lambda_2$ are hyperparameters. $\mathcal{L}_{\text{ort}}$ constrains the subspace and the gradient space of the current LoRA to be orthogonal to those of the previous LoRAs, thereby alleviating forgetting.

### 3.3 Label-guided Localized Balancing Strategy

To model the commonalities of real faces and continuously refine the real face knowledge represented by the backbone in a continual learning sequence, we establish an additional LoRA Expert $\mathbf{A}_0\mathbf{B}_0$, referred to as Real-LoRA, and make it learnable across all tasks. However, the Real-LoRA introduces two issues. First, the additional Real-LoRA disrupts the orthogonality of the LoRA sequence, leading to interference with the Fake-LoRA sequence. Second, it cannot be guaranteed that the Real-LoRA will learn real-face information. To address these issues, we propose a label-guided localized balancing strategy. This strategy dynamically adjusts the responses of experts to different types of samples based on the labels during training. Specifically, it enables Real-LoRA to focus more on real faces, while Fake-LoRAs concentrate on learning forged faces in an orthogonal manner, preventing Real-LoRA from disrupting the orthogonality of Fake-LoRAs. The proposed strategy first takes the inputs from all LoRAs and sets a response matrix $\mathbf{I} \in \mathbb{R}^{(t+1) \times n}$, where $t+1$ indicates the total number of experts. An element $\mathbf{I}_{k,l}$ indicates the response of the $k$-th expert to the $l$-th sample in the batch. It is defined as follows:

$$\mathbf{I}_{k,l} = \sum_{j=1}^{T_m} \frac{\exp\left(\omega_{k,l}^j/\tau\right)}{\sum_{i=0}^{t} \exp\left(\omega_{i,l}^j/\tau\right)}. \tag{10}$$

Here, $\omega_{k,l}^j$ represents the output of the $k$-th expert for the $j$-th token in the $l$-th sample, $T_m$ denotes the number of tokens in the $l$-th sample. All outputs of LoRA can be defined as a vector matrix $\mathbf{O} \in \mathbb{R}^{(t+1) \times n \times d_o}$, where each element is defined as the output of the $k$-th expert for the $l$-th sample. Based on the response matrix $\mathbf{I}$ and the output matrix $\mathbf{O}$, the forward process computes the final output $\mathbf{e} \in \mathbb{R}^{n \times d_o}$, where the output vector $\mathbf{e}_l \in \mathbb{R}^{d_o}$ for the $l$-th sample is defined as:

$$\mathbf{e}_l = \sum_{k=0}^{t} \mathbf{I}_{k,l} \cdot \mathbf{O}_{k,l} \tag{11}$$

To allocate Real-LoRA and Fake-LoRAs for modeling the information of real and fake faces, respectively, we utilize labels to define a balancing coefficient matrix $\mathbf{C} \in \mathbb{R}^{(t+1) \times n}$. An element $\mathbf{C}_{k,l}$ represents the balancing weight applied to the response at the corresponding position $\mathbf{I}_{k,l}$:

$$\mathbf{C}_{k,l} = \begin{cases} 1-\delta, & \text{Type}_e(k) = \text{Type}_h(l) \\ 1+\delta, & \text{Type}_e(k) \neq \text{Type}_h(l) \end{cases}, \tag{12}$$

where $\text{Type}_e(k) = \text{Type}_h(l)$ indicates that the expert matches the category label corresponding to the sample, such as Real-LoRA for real faces, the coefficient matrix $\mathbf{C}$ decreases the balancing coefficient for this item; otherwise, $\mathbf{C}$ increases the balancing coefficient. The increment $\delta$ of the coefficient falls within the range of $[0,1]$. Based on the response matrix $\mathbf{I}$ and the balancing coefficient matrix $\mathbf{C}$, the LLB loss constrains the dispersion of the weighted response matrix $\mathbf{I} \circ \mathbf{C}$:

$$\mathcal{L}_{\text{llb}} = \frac{\sigma^2(\mathbf{I} \circ \mathbf{C})}{\mu(\mathbf{I} \circ \mathbf{C})}, \tag{13}$$

where $\sigma^2(\cdot)$ and $\mu(\cdot)$ represent the variance and mean, respectively, and $\circ$ is element-wise multiplication. This loss function forces the weighted responses ($\mathbf{I} \circ \mathbf{C}$) to become as balanced as possible. By optimizing this loss function, the model is thus compelled to generate a greater original response $\mathbf{I}_{k,l}$ for the matching pairs (to compensate for their small $1 - \delta$ balancing coefficients) and a smaller original response for the mismatched pairs. Additionally, this router does not select specific LoRAs but encourages collaboration among all LoRAs.

When the model begins to learn a new task, DevFD establishes a new LoRA branch and freeze the other Fake-LoRAs in the orthogonal sequence $\{\mathbf{A}_j\mathbf{B}_j\}_{j=1}^{t}$, allowing only the newly added LoRA $\mathbf{A}_t\mathbf{B}_t$ and the Real-LoRA $\mathbf{A}_0\mathbf{B}_0$ to be trainable. During inference, it aggregates the outputs of all LoRA experts to make a joint decision, fostering collaboration among them.

In summary, we define the total loss as the weighted sum of the binary cross-entropy loss $\mathcal{L}_{\text{cls}}$, the orthogonal loss $\mathcal{L}_{\text{ort}}$, and the label-guided localized balancing loss $\mathcal{L}_{\text{llb}}$:

$$\mathcal{L} = \mathcal{L}_{\text{cls}} + \mathcal{L}_{\text{ort}} + \lambda_3 \mathcal{L}_{\text{llb}}, \tag{14}$$

where $\lambda_3$ is a hyperparameter. This loss allows real faces and fake faces from different forgery manipulation types to be modeled by distinct LoRAs without mutual interference and encourages collaboration among all LoRAs in this MoE architecture.

## 4 Experiments

### 4.1 Experimental Setup

**Datasets.** To facilitate comparisons, we select multiple datasets, employing different forgery methods to manipulate and generate faces from distinct domains. FaceForensics++ (FF++) [48] contains 1K real face videos. These faces are manipulated by four forgery methods, resulting in a total of 4K fake videos. The Deepfake Detection (DFD) [12] dataset comprises over a hundred real face samples and more than 1K forged face samples. For the Deepfake Detection Challenge (DFDC) [9] dataset, we utilized the Preview version (DFDC-P), which includes 5K videos and two forgery methods. Celeb-DF v2 (CDF2) [27] comprises 590 real and 5,639 fake videos. DF40 [66] is a large-scale dataset comprising 40 types of forgery methods and over 100K fake videos. We select [FF++, DFDC-P, DFD, CDF2] as our dataset-incremental testing protocol. From DF40, we choose three forgery methods corresponding to different forgery types: Face-Swapping (FS): BlendFace, Face-Reenactment (FR): MCNet, and Entire Face Synthesis (EFS): StyleGAN3. The datasets of selected forgery methods and a base dataset with hybrid forgery types FF++ jointly compose our forgery-type-incremental testing protocol: [Hybrid, FR, FS, EFS].

**Implementation Details and Metrics.** We utilize a pre-trained ViT model as the initial backbone. The pre-training data is sourced from real faces in the FF++ dataset. We conduct self-supervised pre-training through a data augmentation approach [51]. Regarding data utilization, we strictly isolate data across tasks, including real and fake faces. Similar to DFIL [40], we randomly sample 100 videos from the current dataset and train for 20 epochs in each task, maintaining a balance between real and fake samples. The hyperparameter settings are adjusted according to the number of training epochs, and detailed information is provided in Appendix C.2. We employed the Adam optimizer with parameters set to $\beta_1 = 0.9, \beta_2 = 0.999$. The learning rate is set to $1e - 4$, and the batch size is set to 128. The rank (subspace dimension) for LoRA was determined using a grid optimization algorithm similar to InfLoRA [30]. We utilize Accuracy (Acc) and Area Under the Curve (AUC) to evaluate the overall performance of the model. Average Forgetting (AF) indicates the model's ability to retain information from previously learned tasks. Let $a_{t,i}$ denote the evaluation $Score$ (Acc or AUC) on task $i$ after the model has been trained up to task $t$. The $AF_T$ (for $T > 1$) is defined as:

$$AF_T = \frac{1}{T-1} \sum_{i=1}^{T-1} \left( a_{i,i} - a_{T,i} \right), \tag{15}$$

where $a_{T,i}$ is the score on task $i$ after learning the final task $T$.

### 4.2 Comparison Experiments

**Experiments on Dataset Incremental Protocol.** To comprehensively evaluate the performance of the proposed method, we conduct comparative experiments with general continual learning

Table 1: **Left:** Experiments on dataset incremental protocol. **Right:** Experiments on manipulation types incremental protocol. The best performer is highlighted in boldface, while the second-best result is underlined. Shadowed lines indicate the results from our method.

| Dataset Incremental Protocol | | | | | | | | Manipulation Types Incremental Protocol | | | | | | | |
|---|---|---|---|---|---|---|---|---|---|---|---|---|---|---|---|
| Method | Dataset | Acc(%)↑ | | | | Avg↑ | AF↓ | Method | Mani. Type | AUC(%)↑ | | | | Avg↑ | AF↓ |
| | | FF++ | DFDCP | DFD | CDF2 | | | | | Hybrid | FR | FS | EFS | | |
| LWF [28] | FF++ | 95.52 | - | - | - | 95.52 | - | iCaRL [44] | Hybrid | 96.53 | - | - | - | 96.53 | - |
| | DFDCP | 87.83 | 81.57 | - | - | 84.70 | 7.69 | | FR | 67.36 | 99.89 | - | - | 83.63 | 29.17 |
| | DFD | 76.16 | 41.78 | 96.36 | - | 71.43 | 29.58 | | FS | 73.79 | 66.24 | 97.54 | - | 79.19 | 28.20 |
| | CDF2 | 67.34 | 67.43 | 84.05 | 87.90 | 76.68 | 18.21 | | EFS | 52.98 | 55.38 | 64.74 | 100.0 | 68.28 | 40.29 |
| ER [3] | FF++ | 95.52 | - | - | - | 95.52 | - | DER [64] | Hybrid | 97.00 | - | - | - | 97.00 | - |
| | DFDCP | 92.25 | 88.53 | - | - | 90.39 | 3.27 | | FR | 59.03 | 99.73 | - | - | 79.38 | 37.97 |
| | DFD | 84.69 | 80.59 | 94.00 | - | 86.42 | 9.39 | | FS | 68.15 | 19.68 | 97.94 | - | 61.93 | 54.45 |
| | CDF2 | 71.65 | 69.40 | 92.98 | 83.26 | 79.32 | 14.67 | | EFS | 56.79 | 59.83 | 65.36 | 100.0 | 70.49 | 37.56 |
| SI [69] | FF++ | 95.51 | - | - | - | 95.51 | - | CoReD [19] | Hybrid | 96.65 | - | - | - | 96.65 | - |
| | DFDCP | 90.88 | 88.38 | - | - | 89.63 | 4.63 | | FR | 93.55 | 79.88 | - | - | 86.72 | 3.10 |
| | DFD | 45.46 | 27.02 | 96.67 | - | 56.38 | 55.71 | | FS | 89.07 | 79.29 | 86.05 | - | 84.80 | 4.09 |
| | CDF2 | 36.67 | 42.04 | 44.83 | 78.88 | 50.60 | 52.34 | | EFS | 84.54 | 64.29 | 84.17 | 92.63 | 81.41 | 9.86 |
| CoReD [19] | FF++ | 95.50 | - | - | - | 95.50 | - | DFIL [40] | Hybrid | 96.46 | - | - | - | 96.46 | - |
| | DFDCP | 92.94 | 87.61 | - | - | 90.28 | 2.56 | | FR | 55.74 | 99.75 | - | - | 77.75 | 40.72 |
| | DFD | 86.84 | 81.07 | 95.22 | - | 87.71 | 7.60 | | FS | 60.71 | 66.49 | 99.03 | - | 75.41 | 34.51 |
| | CDF2 | 74.08 | 76.59 | 93.41 | 80.78 | 81.22 | 11.42 | | EFS | 50.83 | 95.56 | 70.81 | 99.96 | 79.29 | 26.01 |
| DFIL [40] | FF++ | 95.67 | - | - | - | 95.67 | - | HDP [55] | Hybrid | 96.71 | - | - | - | 96.71 | - |
| | DFDCP | 93.15 | 88.87 | - | - | 91.01 | 2.52 | | FR | 67.41 | 95.45 | - | - | 81.43 | 29.30 |
| | DFD | 90.30 | 85.42 | 94.67 | - | 90.13 | 4.41 | | FS | 63.00 | 71.35 | 95.09 | - | 76.48 | 28.91 |
| | CDF2 | 86.28 | 79.53 | 92.36 | 83.81 | 85.49 | 7.01 | | EFS | 59.89 | 70.06 | 89.34 | 93.73 | 78.26 | 22.65 |
| DMP [56] | FF++ | 95.96 | - | - | - | 95.96 | - | SUR-LID [7] | Hybrid | 96.85 | - | - | - | 96.85 | - |
| | DFDCP | 92.71 | 89.72 | - | - | 91.22 | 3.25 | | FR | 82.91 | 92.42 | - | - | 87.66 | 13.94 |
| | DFD | 92.64 | 86.09 | 94.84 | - | 91.19 | 3.48 | | FS | 90.50 | 96.26 | 97.94 | - | 94.90 | 1.26 |
| | CDF2 | 91.61 | 84.86 | 91.81 | 91.67 | 89.99 | 4.08 | | EFS | 87.90 | 96.79 | 93.56 | 99.07 | 94.33 | 2.99 |
| **DevFD (Ours)** | FF++ | 98.41 | - | - | - | **98.41** | - | **DevFD (Ours)** | Hybrid | 97.63 | - | - | - | **97.63** | - |
| | DFDC-P | 97.06 | 89.90 | - | - | **93.48** | 1.35 | | FR | 94.69 | 93.07 | - | - | **93.88** | 2.94 |
| | DFD | 92.44 | 89.07 | 97.91 | - | **93.14** | 3.40 | | FS | 90.97 | 92.76 | 97.05 | - | 93.59 | 3.49 |
| | CDF2 | 90.71 | 90.31 | 93.12 | 85.15 | 89.82 | 4.03 | | EFS | 90.86 | 94.35 | 95.23 | 99.16 | **94.90** | 2.44 |

approaches (e.g., LwF [28], ER [3], and SI [69]) and specialized continual face forgery detection methods (e.g., CoReD [19], DFIL [40], and DMP [56]). We ensure that the improvements stem from the new strategy and maintain the fairness of the experiments, as detailed in Appendix C.1. Under the dataset incremental protocol, the model is sequentially trained on task sequences constructed from [FF++, DFDC-P, DFD, CDF2]. After completing each task's training phase, we evaluate model accuracy on both the current and previously learned tasks. Quantitative results are organized in a lower triangular accuracy matrix, accompanied by two key metrics: average accuracy (Avg) and average forgetting (AF), as detailed in Table 1-Left. The results reveal that general continual learning methods initially achieve satisfactory accuracy but suffer severe performance degradation on previous tasks as training progresses, resulting in higher forgetting rates. In contrast, continual forgery detection approaches exhibit more stable performance but still suffer from forgetting. It is worth noting that many of the compared methods [7, 40] utilize a replay set, while our method strictly isolates all subtask data and still achieves superior anti-forgetting and state-of-the-art performance.

**Experiments on Manipulation Types Incremental Protocol.** The dataset incremental protocol tested above demonstrates our method's superior performance and anti-forgetting capability. Considering potential issues in the dataset incremental protocol, such as the domain gaps between different datasets and the possibility of shared manipulation methods across datasets, we perform experiments under the manipulation types incremental protocol [Hybrid, FR, FS, EFS], employing AUC as the primary metric for comparison. The experimental results are presented in Table 1-Right. The results show general continual learning approaches (LwF [28], iCaRL [44], DER [64]) exhibit unstable performance with high forgetting rates. Even specialized forgery detection methods demonstrate initial training instability – DFIL [40] shows a 40.72% performance drop on previous tasks during FR-type training. Our method maintains stable learning throughout the entire training sequence, achieving both the highest final performance and the lowest forgetting rate. Comparative analysis reveals significant advantages: our approach surpasses SUR-LID [7] by 0.57% in average AUC, with

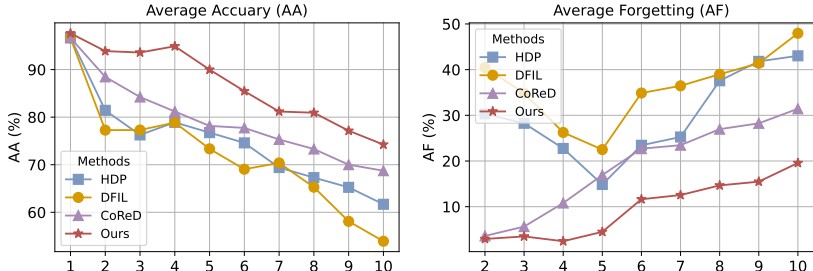

Figure 3: The Task10 long-sequence continual learning experiment based on DF40, the proposed method achieves the highest average accuracy and the lowest forgetting rate.

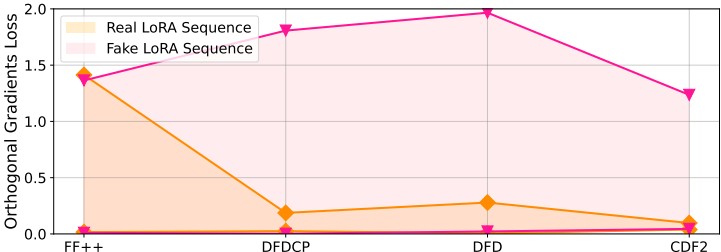

Figure 4: Motivation Validation. We track the variation range of the orthogonal gradient loss during the training process for two orthogonal sequences, one modeling real faces and the other modeling forged faces. For each sequence, the two lines represent the upper and lower bounds of the loss, and the shaded area between them indicates its range of variation. We observe that real faces exhibit an obvious smaller orthogonal gradient loss.

0.55% lower forgetting rate, and outperforms HDP [55] by 16.64% AUC margin while reducing forgetting rate by 20.21%. These results validate our method's performance against manipulation-type shifts in continual learning scenarios.

## 4.3 Long-sequence Continual Learning Experiments

To further validate the effectiveness of our proposed method on more manipulation types and extended task sequences, we selected 10 datasets from DF40 and completed long-sequence continual learning experiments under the Task10 protocol. We provide more details in the Appendix C.3. The average AUC metric and forgetting rate are plotted as shown in Fig. 3. The experimental results indicate that in long-sequence learning, both DFIL [40] and HDP [55] exhibit unstable catastrophic forgetting. At the same time, our method achieves the highest average AUC and the lowest forgetting rates.

## 4.4 Motivation Validation Experiments

Our proposed method is based on a key observation: abundant and unbiased real faces exhibit a more compact distribution across datasets than forged faces created by various methods. Building upon a t-SNE experiment that validates this observation, we further confirm that real faces occupy a more similar input space than forged faces within our proposed framework. We first theoretically demonstrate that a more similar input space corresponds to a smaller orthogonal gradient loss. Specifically, when a new orthogonal LoRA is configured to model a highly similar domain, this domain will possess a highly similar input space $\mathbf{H}_t \in \mathbb{R}^{d_I \times n}$. From Eqn. 7, it can be deduced that the resulting $\mathbf{V}_t^T$ will also be highly similar. Consequently, in Eqn. 8 and Eqn. 9, the term $\sum \|\mathbf{G}_{i,t}\|^2$ yields a near-zero value because its components are approximately orthogonal.

Based on this, we design a new experiment consisting of two separate orthogonal LoRA sequences: one for modeling real faces and the other for modeling forged faces. These sequences learn from the task series [FF++, DFDCP, DFD, CDF2]. We record the range of orthogonal gradient loss values for the two sequences during the training process to investigate the cross-dataset similarity of their respective input spaces. The experimental results are presented in Fig. 4. Starting from the second task, the orthogonal gradient induced by the real face data for a new LoRA is smaller than that induced by the forged face data. The results suggest that real faces from different datasets share a

Table 2: Ablation study on the loss functions. AA represents the average accuracy, while AF indicates the average forgetting.

| Loss | FF++ | DFDC-P | | DFD | | CDF2 | |
|---|---|---|---|---|---|---|---|
| | AA↑ | AA↑ | AF↓ | AA↑ | AF↓ | AA↑ | AF↓ |
| $\mathcal{L}_{cls}$ | 98.18 | 90.49 | 9.33 | 83.86 | 15.79 | 78.82 | 23.93 |
| $\mathcal{L}_{cls} + \mathcal{L}_{olora}$ | 97.30 | 91.08 | 4.53 | 89.71 | 9.01 | 87.66 | 10.06 |
| $\mathcal{L}_{cls} + \mathcal{L}_{ort}$ | 97.35 | 93.59 | 2.68 | 92.08 | 6.82 | **90.66** | 7.91 |
| $\mathcal{L}_{cls} + \mathcal{L}_{llb}$ | 98.09 | **94.20** | 4.45 | 86.75 | 11.19 | 83.07 | 19.31 |
| All | **98.41** | 93.48 | **1.35** | **93.14** | **3.40** | 89.82 | **4.03** |

Table 3: Ablation study on the LoRA modules. The best performer is highlighted in boldface. Shadowed lines indicate the results from our method.

| Module | | FF++ | DFDC-P | | DFD | | CDF2 | |
|---|---|---|---|---|---|---|---|---|
| Real-LoRA Num. | Fake-LoRA Num. | AA↑ | AA↑ | AF↓ | AA↑ | AF↓ | AA↑ | AF↓ |
| 1 | - | 94.02 | 88.94 | 11.44 | 79.05 | 19.51 | 75.69 | 30.80 |
| - | 4 | 97.06 | 93.07 | 2.54 | 91.31 | 4.20 | 88.39 | 6.42 |
| 1 | 4 | **98.41** | **93.48** | 1.35 | **93.14** | **3.40** | 89.82 | 4.03 |
| 4 | 4 | 98.22 | 93.25 | **1.33** | 93.00 | 3.45 | **90.58** | **3.72** |

similar input space $\mathbf{H}_t$, leading to an orthogonal gradient loss that is an order of magnitude smaller. Moreover, the orthogonal LoRA sequence introduces additional computational overhead compared to a single Real-LoRA. This experiment mutually validates our initial t-SNE results, demonstrating that using a single, shared Real-LoRA to model real faces is both reasonable and necessary.

## 4.5 Ablation Study

**Effect of $\mathcal{L}_{ort}$ and $\mathcal{L}_{llb}$.** We employ orthogonal loss $\mathcal{L}_{ort}$ to constrain both the subspace and gradient space of the currently learned LoRA and use $\mathcal{L}_{llb}$ to allocate the Real-LoRA and Fake-LoRAs. We conduct ablation studies on both losses, with the experimental results presented in Table 2. $\mathcal{L}_{olora}$ denotes the use of orthogonal subspaces without orthogonal gradients to validate the effectiveness of our orthogonal gradients; "All" represents the model's performance incorporating all three losses. The results reveal that the $\mathcal{L}_{ort}$ significantly enhance the model's anti-forgetting capability. Meanwhile, $\mathcal{L}_{llb}$ primarily improves the model's Average Accuracy (AA) by facilitating task-specific allocations among LoRAs and maintaining a weighted response matrix for collaboration.

**Effect of Real-LoRA and Fake-LoRAs.** To study the effectiveness of the designed LoRA modules, we separately isolate the two types of LoRA and report the results in Table 3. We differentiate our models based on the number of different types of LoRAs used. Specifically, '4' indicates the use of an orthogonal LoRA sequence (with a sequence length of 4 tasks), '1' indicates the use of a single global LoRA, and '-' indicates that no LoRA module is used. The $\mathcal{L}_{llb}$ is introduced only when both types of LoRAs are used simultaneously. To investigate whether the global Real-LoRA would lead to catastrophic forgetting of real faces, we add an additional experiment shown in the last row by expanding the shared Real-LoRA into a sequence identical to that of Fake-LoRAs, i.e., using two orthogonal LoRA sequences to learn real and fake samples separately. The experimental results show that Fake-LoRAs have a significant effect on mitigating catastrophic forgetting. After incorporating both types of LoRA, the addition of Real-LoRA improves both accuracy and the forgetting rate. Moreover, replacing Real-LoRA with an orthogonal sequence does not bring significant improvements in forgetting rate or accuracy. Therefore, the Real-LoRA does not cause catastrophic forgetting.

## 5 Conclusion

In this paper, we propose DevFD, a Developmental MoE architecture for continual face forgery detection. By utilizing LoRA models as experts, DevFD expands with new LoRA branches to adapt to emerging types of forgery. We employ a label-guided localized balancing strategy to allocate all experts for two purposes: the Real-LoRA refines the real face knowledge modeled by the backbone continuously, while the Fake-LoRAs capture incremental forgery cues from different types. To prevent catastrophic forgetting, we use an orthogonality loss to constrain the learning direction of the learning LoRA to be orthogonal to the existing subspaces. Additionally, we integrate orthogonal gradients into the subspace orthogonal loss to mitigate the interference of gradients on the established subspace knowledge during the entire training phase. Experimental results demonstrate that our approach achieves the best performance.

## Acknowledgement

This work was supported in part by Chinese National Natural Science Foundation Projects U23B2054, 62276254, 62176256, Beijing Natural Science Foundation L242092, the Science and Technology Development Fund of Macau Project 0140/2024/AGJ.

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

# Appendix

## A  Limitations

First, the first limitation lies in the linear increase of our method's model scale as the training sequence progresses. Although the number of parameters in LoRA is very small, this growth may have an impact on extremely long sequence continual learning. Second, our method fine-tunes the pre-trained weights within subspaces using LoRA, and thus may rely on a strong general pre-trained model that already possesses some general knowledge before training. We will address these limitations in our future research.

## B  Details of Orthogonal Gradient

To estimate the gradient space of LoRA and orthogonalize it with the subspace of LoRA, we rely on a key proposition [29] that the gradients of linear layers will lie within the input space. Here, we provide proof of this proposition. For a linear layer, let its weight matrix be denoted as $\mathbf{W} \in \mathbb{R}^{d_I \times d_O}$. Its forward process can be expressed as:

$$\mathbf{e} = \mathbf{W}\mathbf{x} + \mathbf{b}, \tag{16}$$

where $\mathbf{x} \in \mathbb{R}^{d_I}$ and $\mathbf{e} \in \mathbb{R}^{d_O}$ are the input and output vector, respectively. By applying the chain rule, we compute the gradient of $\mathbf{W}$:

$$\frac{\partial \mathcal{L}}{\partial \mathbf{W}} = \frac{\partial \mathcal{L}}{\partial \mathbf{e}} \frac{\partial \mathbf{e}}{\partial \mathbf{W}} = \frac{\partial \mathcal{L}}{\partial \mathbf{e}} \mathbf{x}^T. \tag{17}$$

Assuming the vector $\frac{\partial \mathcal{L}}{\partial \mathbf{e}}$ is given by: $[l_1, l_2, ..., l_{d_O}]^T$, then the gradient $\frac{\partial \mathcal{L}}{\partial \mathbf{e}} \mathbf{x}^T$ can be represented as:

$$\frac{\partial \mathcal{L}}{\partial \mathbf{e}} \mathbf{x}^T = \begin{bmatrix} l_1, \\ l_2, \\ ..., \\ l_{d_O} \end{bmatrix} \mathbf{x}^T = \begin{bmatrix} l_1 \mathbf{x}^T, \\ l_2 \mathbf{x}^T, \\ ..., \\ l_{d_O} \mathbf{x}^T \end{bmatrix}. \tag{18}$$

It can be observed that each row of the gradient $\frac{\partial \mathcal{L}}{\partial \mathbf{W}}$ is a multiplication of the input vector $\mathbf{x}$ with a certain value $l_k$ ($1 \leq k \leq d_O$). Hence, the gradient shares the same subspace as the input vector $\mathbf{x}$, meaning the gradient lies within the input space.

## C  More Experimental Details

### C.1  Experimental Fairness

First, at the data level, we use the same amount of subtask data as the compared methods, i.e., 100 videos with 20 sampled frames per video, which is identical to methods such as DFIL [40] and SUR-LID [7]. Second, at the model level, we employ a backbone with a similar order of magnitude of parameters as the compared methods: ViT-B/16, thus avoiding the use of a larger model to ensure experimental fairness. It is worth noting that our trainable parameters consist only of the parameters of two LoRAs, which is approximately 1/10 of the trainable parameters of the compared methods. Third, in terms of training strategies, many methods such as DFIL [40] and SUR-LID [7] utilize a Replay Set, which allows access to data from previous tasks—a practice that is prohibited in many continual learning scenarios. The proposed DevFD method completely isolates any data between subtasks. This not only demonstrates that we achieve the best results under stricter constraints but also indicates the broader applicability of our method. Finally, the comparative metric data used are the best results extracted directly from the original papers. We achieve an undeniably leading position in absolute performance.

### C.2  More Implementation Details

We provide our detailed hyperparameter settings here. For the label-guided localized balancing strategy, we use a fixed hyperparameter setting. We set $\delta$ to 0.15 and $\lambda_3$ to 0.2. For the integrated

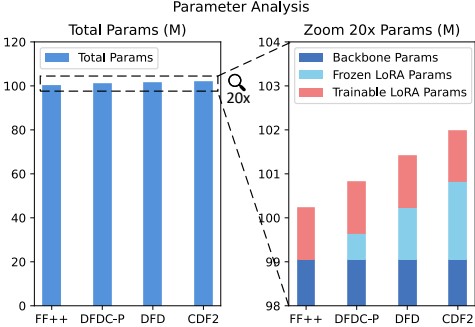

Figure 5: Expanded parameter analysis. The parameter expansion of the proposed developmental MoE is fully controllable, and the trainable parameters will remain constant throughout the task sequence.

orthogonality loss, since our goal is to prevent the gradients in the current LoRA from interfering with the subspaces established by previous tasks during the entire training phase, we dynamically adjust the hyperparameters of the orthogonal subspace loss to the orthogonal gradient loss for each epoch. Specifically, from epoch [0,5), we set $\lambda_1$ to 0.5 and $\lambda_2$ to 0.5. From epoch [5,10), we set $\lambda_1$ to 1 and $\lambda_2$ to 0.1. From epoch [10,20), we set $\lambda_1$ to 1 and $\lambda_2$ to 0.01.

### C.3 More Details of Task10 Long Sequence Experiment

Existing continual learning methods for forgery detection typically employ task sequences with only four subtasks. We provide results for a continual learning task sequence with 10 subtasks based on DF40 [66] and reproduce three high-performing continual learning methods for comparison: CoReD [19], DFIL [40], and HDP [55]. We start with the original task sequence [Hybrid, FR, FS, EFS], where Face-Swapping (FS): BlendFace, Face-Reenactment (FR): MCNet, and Entire Face Synthesis (EFS): StyleGAN3 are used as our initial learning sequence and expand it to 10 tasks. We carefully select the remaining six methods while maintaining diversity. Specifically, for Face-Swapping (FS), we choose SimSwap and InSwapper; for Face-Reenactment (FR), we select Wav2Lip and SadTalker; and for Entire Face Synthesis (EFS), we choose StyleGAN2 and StyleGAN-XL. We then randomly shuffle the last six methods to form the complete task sequence with 10 subtasks: [FFpp, MCNet, BlendFace, StyleGAN3, StyleGAN-XL, Wav2Lip, SadTalker, StyleGAN2, SimSwap, InSwapper]. Analysis reveals that EFS methods may have significant differences from FS and FR methods, leading to sharp increases in forgetting rates during transitions between learning EFS methods and FS & FR methods, such as 5->6 and 7->8.

## D More Experimental Results

### D.1 Parameter Analysis

We introduce a Developmental Mixture of Experts: DevFD to tackle the challenge of detecting new types of face forgeries. As new tasks are integrated, the parameters of the DevFD model expand in a specific manner. To illustrate that our developmental MoE can handle long-sequence tasks without suffering from parameter explosion, we perform experiments to track the expansion in parameter volume, as depicted in Fig. 5. By employing Parameter-Efficient Fine-Tuning (PEFT) technology, our DevFD model can learn in long-sequence tasks with a minimal increase in parameter volume. The experimental results show that when the model is fine-tuned for the first task in the sequence, the trainable parameters constitute only 1.18% of the total parameters. Even after training on all four tasks, the additional parameters account for just 2.89% of the total. Furthermore, since only the Real-LoRA and the final LoRA in the Fake-LoRA sequence are trainable, the number of trainable parameters remains stable throughout the task sequence and does not expand with the overall parameter volume. Consequently, the parameters of our proposed developmental MoE are fully manageable, making it well-suited for learning in long-sequence tasks.

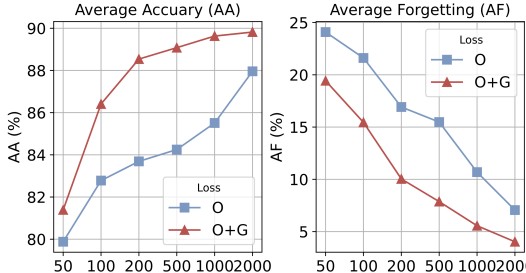

Figure 6: Effect of subspace and gradient constraint. O represents the scenario with only subspace constraints, while O+G indicates the presence of both subspace and gradient constraints. The gradient constraints improve in both accuracy and forgetting rate.

## D.2  Effect of orthogonal Subspace and Orthogonal Gradients

The proposed method introduces additional gradient constraints to ensure that even when the subspaces are not entirely orthogonal, the gradient space remains orthogonal to the established subspaces. This prevents the gradient from interfering with the established subspaces. We simulate the early stages of the training process using fewer training samples to evaluate whether these constraints on the gradient space can prevent interference and the results shown in Fig. 6. Here, O represents the scenario with only subspace constraints, while O+G indicates the presence of both subspace and gradient space constraints. The experimental results demonstrate that the orthogonal gradients significantly reduces the average forgetting in the early stages of training. Additionally, it can improve the average accuracy by constraining the early learning direction, confirming the effectiveness of orthogonal gradients.

## D.3  Effect of ViT Backbone

We employ a Vision Transformer (ViT) pre-trained by CLIP [43] as our initial backbone for training. To explore how different backbones affect model performance, we test various ViT models [16] with varying sizes of parameters. The results of these experiments are detailed in Table 4.

Table 4: Effect of backbones in the proposed method.Shadowed lines indicate the backbone used in our method.

| Backbone | Params | FF++ | DFDC-P | | DFD | | CDF2 | |
|---|---|---|---|---|---|---|---|---|
| | | AA | AA | AF | AA | AF | AA | AF |
| ViT-B/16 | 86M | 98.41 | 93.48 | 1.35 | 93.14 | 3.40 | 89.82 | 4.03 |
| ViT-B/32 | 86M | 98.03 | 91.24 | 2.88 | 93.29 | 4.52 | 88.25 | 7.63 |
| ViT-L/14 | 304M | 99.07 | 93.75 | 1.36 | 93.37 | 3.79 | 90.55 | 5.86 |

Our experiments reveal that a patch size of 16x16 performs better than 32x32, with the ViT-B/16 model demonstrating superior overall performance compared to the ViT-B/32 model. When considering the effect of increasing the parameter scale, the ViT-L/14 model, which adds 218M parameters to the

Table 5: Performance comparison of different task orders.

| Dataset | FF++ | DFDC-P | CDF2 | DFD | AA | AF |
|---|---|---|---|---|---|---|
| FF++ | 98.41 | - | - | - | 98.41 | - |
| DFDC-P | 96.95 | 90.88 | - | - | 93.92 | 1.46 |
| CDF2 | 93.42 | 88.37 | 92.33 | - | 91.37 | 3.75 |
| DFD | 90.86 | 84.35 | 89.81 | 95.91 | 90.23 | 4.15 |
| Dataset | FF++ | DFD | DFDC-P | CDF2 | AA | AF |
| FF++ | 98.41 | - | - | - | 98.41 | - |
| DFD | 94.13 | 97.25 | - | - | 95.69 | 4.28 |
| DFDC-P | 92.88 | 94.91 | 85.94 | - | 91.24 | 3.94 |
| CDF2 | 90.05 | 94.39 | 87.28 | 90.43 | 90.54 | 3.29 |

ViT-B/16, only saw a modest improvement in average accuracy but a slight increase in the average forgetting rate when trained on all four tasks. We attribute this marginal gain to the limited dataset size, which makes model performance less sensitive to parameter scale increase. Based on these findings, we choose the ViT-B/16 as our backbone model for its optimal balance of performance and efficiency.

### D.4  Effect of Task Orders

To determine if the high accuracy and resistance to forgetting in our proposed method are independent of the specific order in which tasks are presented, we conducted a series of experiments varying the task sequences. The outcomes of these experiments are detailed in Table 5. Our findings indicate that our method consistently delivers high detection accuracy and resistance to forgetting across different task sequences. For instance, when the tasks are sequenced as [FF++, DFDC-P, CDF2, DFD], the model achieves an average accuracy of 90.23% and an average forgetting rate of 4.15%. In another sequence, [FF++, DFD, DFDC-P, CDF2], the model's average accuracy is 90.54% with an average forgetting rate of 3.29%. These results suggest that the order of tasks does not significantly impact the model's performance.

