# OpenReview forum: "DevFD : Developmental Face Forgery Detection by Learning Shared and Orthogonal LoRA Subspaces"
_NeurIPS.cc/2025/Conference — NeurIPS 2025 poster_

### Official Review · Reviewer_1J7u · 2025-06-29

**Clarity:** 3
**Significance:** 2
**Originality:** 2
**Rating:** 3
**Confidence:** 5

**Summary:**

This work proposes to address the challenge of continual face forgery detection. The authors assume that the real faces are more stable and singular compared to fake ones. The proposed framework incorporates LoRA-based experts split into two groups: one for refining real face features and others for learning different forgery types incrementally. To avoid catastrophic forgetting, Fake-LoRAs are trained with orthogonal constraints on their learning directions. Experiments conducted on various datasets and forgery types show the method’s effectiveness.

**Questions:**

See weaknesses.

**Ethical Concerns:**

["NO or VERY MINOR ethics concerns only"]

**Final Justification:**

I appreciate the authors’ effort and time in providing detailed explanations and justifications. However, my major concerns regarding the novelty, the underlying assumptions, and the contributions to the community remain unaddressed.  I concur with reviewers gTcD and BQKM that the use of MoE for continual learning, deepfake detection, and AIGC detection has been extensively explored in prior work. Although the authors attempt to highlight the differences between their approach and existing methods, I still find this work to be a relatively straightforward combination of existing techniques applied to the task of continual face forgery detection.  Furthermore, the authors have not provided additional experiments to validate their strong assumption that real faces (regardless of source or domain) are stable and can be modeled by a single Real-LoRA.  I also find that the method lacks task-specific innovations for face forgery detection, resulting in limited contributions to the community. While the proposed label-guided localized balancing strategy is interesting, it is a general technique that could be applied to some other binary classification tasks and is not uniquely tailored to face forgery detection. Given these concerns, I am inclined to keep my rating unchanged.

I carefully reviewed the authors' response and the manuscript. The additional experiments partially addressed my concerns. However, I still believe the quality of this work is below the standard of NeurIPS.

**Limitations:**

Yes

**Quality:**

2

**Strengths And Weaknesses:**

Strengths
- The paper is well-structured and easy to follow.

- The dataset-incremental and manipulation-incremental results are promising, and the ablation studies well demonstrate the effectiveness of the designed components.

Weaknesses
- The novelty of this work appears to be limited. Adapting MoE to continual learning has been studied in [A], and the MoE learning scheme for  face forgery detection has been explored in [B]. Moreover, applying SVD for deepfake detection has been previously studied [C].

- This work is based on a strong assumption that real faces are stable and can be modeled by one single Real-LoRA, which appears to be a key contribution of this work. However, this assumption is only supported by one t-SNE map of two datasets. More preliminary studies are expected to supprot this assumption.

- Although the proposed method demonstrates good performance on continual face forgery detection benchmarks, the model architecture lacks design specifically tailored to the task of face forgery detection. As such, the work offers limited insight into the reasons behind the achieved performance gains.

- The experiments are conducted on four datasets, including FF++ and three Deepfake datasets: DFDCP, DFD, and CDF2. Additional experiments focusing on continual learning across unseen manipulation types would better demonstrate the generalizability of the proposed method. For manipulation-incremental experiments, it is unclear why the authors only pick three manipulation types from DF40.

[A] Boosting continual learning of vision-language models via mixture-of-experts adapters, CVPR, 2024.

[B] Moe-ffd: Mixture of experts for generalized and parameter-efficient face forgery detection, ArXiv, 2024.

[C] Orthogonal Subspace Decomposition for Generalizable AI-Generated Image Detection, ICML 2025.

---

> ### Author Rebuttal · Authors · 2025-07-30
>
> **Thank you for your thoughtful and constructive review. Our responses to your questions are as follows:**
> ### W1: The novelty of our contribution in contrast to prior works [A, B, C].
>
>    We will now elaborate on the differences between our method and three related approaches:
>
>    1. **[A]** employs conventional MoE Adapters for Class-Incremental Learning. First, it uses a traditional router with a top-k mechanism to select experts, which necessitates T separate routers for T tasks. In contrast, our method utilizes a soft routing mechanism, which reduces the parameter count and emphasizes cooperation among experts. Second, MoECL's activate-freeze strategy can only preserve knowledge by freezing experts; it cannot prevent the experts being trained on the current task from interfering with the knowledge acquired by previous experts. Third, [A] is intended for general-purpose continual learning, while our method is specifically designed for the characteristics of the face forgery detection task and includes a mechanism to balance the experts, significantly improving model performance.
>    2. **[B] is an MoE framework for static models.** The experts in this MoE are pre-defined, whereas ours are dynamically expanded. Consequently, this method cannot adapt to continual learning scenarios that involve various emerging forgery types.
>    3. **[C]** utilizes SVD to decompose CLIP weights, with the intent of preserving CLIP's general-purpose knowledge. **In our work, SVD serves as a dimensionality reduction operation to project the orthogonal gradient into the LoRA subspace**. Thus, the methodological objective and the implementation path are fundamentally different. Furthermore, [C] is also a static method and is not applicable to continual learning scenarios.
>
>    To the best of our knowledge, our work is the first to apply a combination of MoE and orthogonal gradients to the task of continual face forgery detection.
>
> ### W2: The assumption of modeling real faces with a single, stable distribution.
>
>    This is a **key observation** that serves as a motivation for our work: real faces tend to be more compactly distributed in the feature space, while different types of forged faces exhibit greater variance among themselves. **This makes it reasonable to model real faces and different types of forgeries with separate, dedicated LoRA modules.** The final results validate the effectiveness of our method. We will add more detailed experiments with different backbones in the main text to further substantiate this point.
>
> ### W3: On how our architectural design is specifically tailored to the face forgery detection task.
>
>    We must emphasize our **label-guided localized balancing strategy**, which is specifically designed for continual face forgery detection and has not been explored in papers on general-purpose continual learning. We leverage the key insight that authentic faces exhibit high similarity across datasets, whereas forged faces have significant distributional gaps arising from different manipulation techniques. Our strategy uses this prior to softly assign a shared "Real-LoRA" to model the common features of authentic faces, and a sequence of orthogonal LoRAs to model different forgery types independently and without interference.
>
>    As demonstrated  in Section 4.3, titled "Effect of Lort and Lllb," and in Table 2, the third row presents the results of using only the orthogonal loss without the label-guided localized balancing strategy. The results indicate that the label-guided localized balancing strategy reduces the forgetting rate by 3.88% (from 7.91 to 4.03), significantly improving model performance.
>
> ### W4: Additional experiments focusing on continual learning across unseen manipulation types.
>
>    We did not limit our selection to only three forgery methods from DF40. On the contrary, **we conducted more extensive, long-sequence experiments on this dataset, as shown in Figure 3.** We selected 10 forgery types from DF40, ensuring that our selection covers all major forgery categories as evenly as possible. Using these 10 selected types, we conducted a standard 10-task long-sequence continual learning experiment. The results demonstrate that our method achieves the best performance and the lowest forgetting rate, even in long-sequence scenarios and when evaluated against multiple unseen manipulation types.

---

> ### Author Response · Authors · 2025-08-05
>
> Dear Reviewer 1J7u,
>
> We sincerely thank you for your insightful and constructive review. In our rebuttal, we have provided detailed explanations for your concerns regarding the novelty of our work, our underlying assumptions, and the experiments on long-sequence continual learning. We hope that these responses have successfully addressed your questions. We look forward to your feedback and welcome any further discussion.
>
> Thank you again for your valuable time and expert review.
>
> Yours Sincerely,
> Authors

---

> > ### Comment · Reviewer_1J7u · 2025-08-06
> >
> > I appreciate the authors’ effort and time in providing detailed explanations and justifications. However, my major concerns regarding the novelty, the underlying assumptions, and the contributions to the community remain unaddressed.  I concur with reviewers gTcD and BQKM that the use of MoE for continual learning, deepfake detection, and AIGC detection has been extensively explored in prior work. Although the authors attempt to highlight the differences between their approach and existing methods, I still find this work to be a relatively straightforward combination of existing techniques applied to the task of continual face forgery detection.  Furthermore, the authors have not provided additional experiments to validate their strong assumption that real faces (regardless of source or domain) are stable and can be modeled by a single Real-LoRA.  I also find that the method lacks task-specific innovations for face forgery detection, resulting in limited contributions to the community. While the proposed label-guided localized balancing strategy is interesting, it is a general technique that could be applied to some other binary classification tasks and is not uniquely tailored to face forgery detection. Given these concerns, I am inclined to keep my rating unchanged.

---

> > > ### Author Response · Authors · 2025-08-07
> > >
> > > Dear Reviewer 1J7u,
> > >
> > > Thank you for your insightful comments, which are helping us improve our paper. We sincerely apologize that our rebuttal did not resolve your major concerns. However, we have since made new experimental progress specifically addressing the **underlying assumptions** and the **novelty** of our work. These new experimental results directly target the two main concerns you have raised. We would like to share this progress with you and hope that it resolves your concerns.
> > >
> > > Regarding the **underlying assumptions**, we were initially inspired by a t-SNE experiment, which suggested that abundant and unbiased real faces exhibit a more compact distribution across datasets compared to forged faces created by various methods. **This inspirational hypothesis has now been further validated by a new form of experiment.** As we reached a consensus with Reviewer soP2 in our discussion, within a sequence of orthogonal LoRA modules, **a new LoRA models knowledge that is complementary to all preceding LoRAs** under the constraints of orthogonal gradients and subspaces. When a new orthogonal LoRA is configured to model a repeated domain, ***this domain will have an identical or highly similar input space, $H_{t} \in \mathbf R^{d_{I} \times n}$. From Equation 7, it can be deduced that the resulting $(V_t)\_{r}$ will also be highly similar. Consequently, in the orthogonal gradient calculation (Equation 9), the term $\\sum \\Vert (V\_{t})\_{r} B\_{i} \\Vert^{2}$ will yield a near-zero gradient because its components are already approximately orthogonal.*** This leads to a small orthogonal gradient loss that is difficult to minimize, making it challenging for the new LoRA to learn from the repeated domain. A more detailed description of this phenomenon can be found in our discussion with Reviewer soP2.
> > >
> > > Based on this, we designed a new experiment composed of two separate orthogonal LoRA sequences: **one for modeling real faces and another for modeling forged faces.** These sequences learn from the task series [FF++, DFDCP, DFD, CDF2], and we recorded the **approximate values of the orthogonal gradient loss for both sequences** to investigate the cross-dataset similarity of the input spaces. The experimental results are presented in the table below:
> > >
> > > | Orthogonal Gradients Loss                  | FF++           | DFDCP          | DFD            | CDF2           |
> > > | ------------------------------------------ | -------------- | -------------- | -------------- | -------------- |
> > > | **Orthogonal LoRA Sequence for Real Face** | 1.413 to 0.016 | 0.187 to 0.025 | 0.279 to 0.005 | 0.097 to 0.039 |
> > > | **Orthogonal LoRA Sequence for Fake Face** | 1.364 to 0.010 | 1.807 to 0.002 | 1.966 to 0.022 | 1.236 to 0.045 |
> > >
> > > We observed that, starting from the second task, **the orthogonal gradient induced by the real face data for a new LoRA was approximately $\frac{1}{10}$ of that induced by the forged face data.** We also observed a slower decrease in this loss, indicating that it is more difficult for the new orthogonal LoRA to learn from the real faces in subsequent tasks. The results suggest that **real faces from different datasets share a similar input space $H_{t} \in \mathbf R^{d_{I} \times n}$**, leading to an orthogonal gradient loss that is an order of magnitude smaller. Moreover, the preceding LoRA modules introduce additional computational overhead compared to a single Real-LoRA. **This experiment mutually validates our initial t-SNE results, demonstrating that using a single, shared Real-LoRA to model real faces is both reasonable and necessary.** Due to time constraints, we will incorporate a detailed analysis and quantitative metrics of this experiment into the final version of the paper.

---

> > > > ### Author Response · Authors · 2025-08-07
> > > >
> > > > Regarding the **novelty** of our method, the analysis above demonstrates the unique properties of face forgery detection: **real faces possess substantial common knowledge across datasets, whereas forged faces exhibit complementary knowledge.** Compared to methods like DFIL, which can be directly applied to binary classification in a continual learning setting, **our approach is designed specifically around these unique data distribution properties of the face forgery detection task, making it both innovative and highly task-specific.** Here, we will further elaborate on the novelty of our work. We illustrate this with an experiment that we began during the rebuttal phase but were unable to include previously due to time limits. This experiment shows that **general-purpose continual learning methods using MoE and LoRA do not perform well on the continual face forgery detection task, whereas our proposed strategy of using face detection distribution characteristics to guide different LoRA modules can significantly improve performance.** Specifically, we applied the general-purpose continual learning methods, InfLoRA, MoECL and O-LoRA (which was included in the comparisons in our paper), to the continual face forgery detection task and compared them with our DevFD, as shown in the table below:
> > > >
> > > > | Datasets        | FF++      | DFDCP     | DFDCP    | DFD       | DFD      | CDF2      | CDF2     |
> > > > | --------------- | --------- | --------- | -------- | --------- | -------- | --------- | -------- |
> > > > | Metrics         | AA        | AA        | AF       | AA        | AF       | AA        | AF       |
> > > > | MoECL           | 98.13     | 90.71     | 6.54     | 89.04     | 9.13     | 83.74     | 12.67    |
> > > > | O-LoRA          | 97.30     | 91.08     | 4.53     | 89.71     | 9.01     | 87.66     | 10.06    |
> > > > | InfLoRA         | 97.86     | 90.19     | 5.33     | 90.37     | 5.27     | 85.44     | 7.05     |
> > > > | **Ours(DevFD)** | **98.41** | **93.48** | **1.35** | **93.00** | **3.45** | **90.58** | **3.72** |
> > > >
> > > > We found that, lacking designs tailored to the specific characteristics of the continual forgery detection task, these **general-purpose SOTA methods fail to achieve satisfactory results when directly applied**. **Our method, designed specifically for the distributional properties of real and forged faces, achieves performance far superior to these general SOTA continual learning methods.** We believe this strongly demonstrates the **novelty** and **task-specific effectiveness** of our approach for continual face forgery detection. We will also incorporate this experiment into the main paper.
> > > >
> > > > Thank you once again for your insightful comments.
> > > >
> > > > Sincerely,
> > > >
> > > > The Authors

---

### Official Review · Reviewer_BQKM · 2025-07-01

**Clarity:** 4
**Significance:** 4
**Originality:** 4
**Rating:** 4
**Confidence:** 5

**Summary:**

This paper proposes DevFD, a Developmental Mixture of Experts (MoE) architecture designed for continual face forgery detection. Addressing the challenge of rapidly evolving forgery techniques, DevFD frames face forgery detection as a continual learning problem. The core of the method involves utilizing Low-Rank Adaptation (LoRA) models as individual experts, allocating them into two groups: a Real-LoRA to refine the understanding of genuine facial features, and a sequence of Fake-LoRAs to incrementally capture information from various, emerging forgery types. A key innovation is the integration of orthogonal gradients into the orthogonal loss for Fake-LoRAs, aiming to prevent catastrophic forgetting by ensuring that learning new forgery types does not interfere with established knowledge. Furthermore, a label-guided localized balancing strategy is introduced to appropriately allocate expert responses during training.

**Questions:**

See the weakness.

**Ethical Concerns:**

["NO or VERY MINOR ethics concerns only"]

**Final Justification:**

The author's rebuttal addresses some of my questions, but the generalization to specific environments is not well demonstrated experimentally. Therefore, I maintain my recommendation of "borderline reject".

**Limitations:**

Yes

**Quality:**

4

**Strengths And Weaknesses:**

1. The paper introduces a well-conceived Developmental MoE architecture specifically tailored for the unique challenges of continual face forgery detection, where real faces are stable and abundant, while fake faces continuously evolve. This frames the problem effectively within a continual learning paradigm.
2. The theoretical analysis identifying the limitations of pure subspace orthogonality for preventing forgetting, and the subsequent proposal of integrating orthogonal gradients into the orthogonal loss (L_ort), is a significant technical contribution. This mechanism directly addresses the issue of early training interference, crucial for robust continual learning.
3 .The design to explicitly model real faces via a dedicated Real-LoRA and evolving fake faces via orthogonal Fake-LoRAs, coupled with the label-guided localized balancing strategy (L_llb), allows for efficient and targeted knowledge acquisition without mutual interference, while also fostering overall expert collaboration.
4.DevFD demonstrates state-of-the-art average accuracy and, critically, the lowest average forgetting rates across comprehensive experiments on both dataset-incremental and manipulation-type-incremental protocols, including challenging long-sequence tasks. This provides robust evidence of its practical effectiveness. The utilization of LoRA modules as experts ensures that the model grows in a highly parameter-efficient manner as new tasks are learned, keeping the number of trainable parameters minimal compared to the backbone, which is important for scalability in long-sequence learning.

Weakness:

1. While the orthogonal gradient contribution is somehow novel, the fundamental concept of using LoRA with MoE for continual learning has been explored in recent works like MoECL, O-LoRA, and InfLoRA. The paper could articulate more explicitly how DevFD’s architectural design and mechanisms fundamentally differ from these existing methods, beyond just the gradient constraint, especially in how it leverages the unique asymmetry of real vs. fake data in FFD.

2 .Although the paper highlights parameter efficiency, it lacks a comprehensive analysis of the computational cost (e.g., training time per task, inference time per image) compared to baseline methods. The added complexity of computing and enforcing orthogonal gradients, and the overall MoE routing, warrants a discussion on the trade-off between performance gain and computational overhead, which is critical for real-world deployment.

3.The method introduces several hyperparameters (e.g.,delta,lambda3,and the dynamically adjusted lambda1, lambda2). While settings are provided, a more in-depth analysis of their sensitivity and how robustly these optimal values were determined across different tasks or initial conditions would strengthen the paper.

4.While LoRA parameters are small, the long-term implications for extremely prolonged continual learning sequences (e.g., 50+ tasks) regarding total model size and memory footprint are not fully explored.

---

> ### Author Rebuttal · Authors · 2025-07-30
>
> **Thank you for your thoughtful and constructive review. Our responses to your questions are as follows:**
> ### W1: Clarification on our architectural novelty for the forgery detection task, in contrast to general-purpose continual learning methods.
>    Yes, we appreciate your suggestion. First, we will elaborate on the key distinctions between our method and MoECL, O-LoRA, and InfLoRA:
>
>    1. **MoECL** employs conventional MoE Adapters for Class-Incremental Learning. It uses a traditional router with a top-k mechanism to select experts, which necessitates T separate routers for T tasks. In contrast, our method utilizes a soft routing mechanism, which reduces the parameter count and emphasizes cooperation among experts. Furthermore, MoECL's activate-freeze strategy can only preserve knowledge by freezing experts; it cannot prevent the experts being trained on the current task from interfering with the knowledge acquired by previous experts.
>    2. **O-LoRA** first proposed using subspace orthogonality to prevent catastrophic forgetting. However, it enforces this orthogonality via a loss function. During the training process, when this loss value is large and the orthogonality condition is not yet met, the learning process can still disrupt previously acquired knowledge. Our approach mitigates this by orthogonalizing the gradients themselves, preventing such interference throughout the entire training process.
>    3. **InfLoRA** uses a pre-defined dimensionality reduction matrix, Bt, which restricts the learning capacity of LoRA as an adapter. In contrast, our use of orthogonal gradients and subspaces ensures that LoRA can effectively adapt to new tasks while simultaneously preventing catastrophic forgetting.
>
>    Second, we wish to emphasize our **label-guided localized balancing strategy**, which is specifically designed for continual face forgery detection—an aspect unexplored by the aforementioned methods. We leverage the key insight that authentic faces exhibit high similarity across datasets, while forged faces have significant distributional gaps due to different manipulation techniques. Our strategy uses this prior to softly assign a shared "Real-LoRA" to model authentic faces, and a sequence of orthogonal LoRAs to model different forgery types independently and without interference.
>
>    Third, we provide experimental evidence for our method's effectiveness. In **Appendix C.4**, we present the results of our framework using only the orthogonal subspace constraint (the core mechanism of O-LoRA). The results demonstrate that our complete method significantly outperforms O-LoRA.
>
> ### W2: Analysis of the computational cost (e.g., training time per task, inference time per image) compared to baseline methods.
>
>    Thank you for your attention to our model's efficiency. We provide the details of the computational cost below.
>
>    | Method              | training time per batch | inference time per dataset（1000 images） |
>    | :------------------ | ----------------------- | ----------------------------------------- |
>    | Baseline | 9.93                    | 10.23                                     |
>    | DevFD (Ours)        | 9.94                    | 10.23                                     |
>
>    1. First, regarding the training cost, we use two A6000 GPUs with a batch size of 64. **For a typical task, the training time per batch is 9.94 seconds**. **Without the orthogonal space and orthogonal gradient computation, the time is 9.93 seconds, which is nearly identical.** Furthermore, in our training process, the number of learnable parameters remains constant as tasks increase. This means that an increase in the number of tasks in the sequence does not significantly extend this training time.
>
>    2. Regarding inference, the **pure inference time** for a typical dataset of approximately 1,000 images at a 224x224 resolution is about 10 seconds. This time also shows **negligible increase as the training sequence lengthens.** The reason is that the added computational load from the multiplication of the two low-rank matrices in each new LoRA expert is minimal.
>
>    3. Furthermore, we will add a detailed analysis of the computational cost to the main paper. This analysis will include an ablation study, where we progressively remove components of our model to investigate the trade-off between performance gains and computational cost.
>
> ### W3: Analysis of hyperparameters sensitivity.
>
>    Thank you for your suggestion on refining our method. We present an ablation study for the different loss components in **Table 2**, where each loss is removed by setting its corresponding hyperparameter to zero. Furthermore, we will add a comprehensive experiment on hyperparameter values and their sensitivity to the main text of the paper.
>
> ### W4:  Total model size and memory footprint.
>
>    We provide a parameter count analysis for sequential learning in **Appendix C.7.** The number of our training-related learnable parameters remains constant throughout the task sequence, without increasing as new tasks are added. This number is very small, **accounting for only 1.18% of the backbone's parameters**. After completing all four tasks, our total parameter **count increases by only 2.89% compared to the initial state**. This growth in parameters is almost negligible.

---

> ### Comment · Reviewer_BQKM · 2025-08-05
>
> Thank you for your detailed rebuttal. Currently, I prefer to keep my rating of "4: Borderline accept" based on considering the overall contribution of this work.

---

### Official Review · Reviewer_soP2 · 2025-07-04

**Clarity:** 4
**Significance:** 3
**Originality:** 3
**Rating:** 4
**Confidence:** 4

**Summary:**

The paper aims to tackle the ever-evolving new types of forgery issues. It frame face forgery detection as a continual learning problem and allow the model to scale in complexity as new forgery types emerge. Specifically,  it employs a Developmental Mixture of Experts (MoE) architecture, utilizing LoRA models as the individual experts. To prevent catastrophic forgetting, the paper ensures that the learning direction of Fake-LoRAs is orthogonal to the established subspace.

**Questions:**

The authors should clarify whether extending LoRA modules specifically for the real data branch could further enhance performance, and provide a more thorough analysis of the current routing distribution.

**Ethical Concerns:**

["NO or VERY MINOR ethics concerns only"]

**Final Justification:**

The author's explanation has satisfactorily addressed my main concerns. Therefore, I have decided to change my rating to borderline accept.

**Limitations:**

The authors should clearly state the limitations of their method, such as which types of attacks it can address and which types it cannot.

**Quality:**

3

**Strengths And Weaknesses:**

Strengths
1. The motivation of the paper is clear, and the proposed solution is straightforward.

Weaknesses
1. How do you define an "expert"? In practice, it is often difficult to accurately determine the specific attack method for an attack image. Would such hybrid or ambiguous attack types affect the performance of the current MoE (Mixture of Experts) framework?
2. Since real data can have a wide distribution and may include new scenarios (e.g., images from different cameras, people of different ethnicities, etc.), would introducing an additional adapter specifically for real data further improve performance?
3. The paper lacks an in-depth analysis of the routing distribution. Is there a difference in routing behavior for different types of attacks?
4. Why does the proposed method outperform previous approaches even on a first dataset (for example, achieving 98.41% on FF++)?
5. The method itself is essentially an extension using multiple LoRA modules, which may not be particularly novel.

---

> ### Author Rebuttal · Authors · 2025-07-30
>
> **Thank you for your thoughtful and constructive review. Our responses to your questions are as follows:**
> ### W1: Definition of "expert" and the framework's robustness against ambiguous or hybrid attack types.
>
>    Thank you for your insightful feedback. We acknowledge that precisely defining specific attack types is challenging in practice. Our model is designed with this consideration in mind, and its performance is **not adversely affected by hybrid or ambiguous attack types.** We explain the reasons from both the training and testing perspectives below:
>
>    1. **During the training phase:** The training data typically includes known dataset labels and forgery type labels. We leverage this information to extend the MoE framework, allowing one expert to correspond to a specific dataset or forgery type. In fact, our experiments on the FF++ dataset, which is a composite of four different manipulation methods, support this. In our dataset-incremental protocol, we use a single LoRA to model the forged data from this mixed-type dataset and still achieve state-of-the-art results. This outcome empirically demonstrates that our framework is robust to hybrid or ambiguous attacks.
>    2. **During the testing or inference phase:** At this stage, dataset and type labels are unavailable. Our framework does not require the forgery type to be known. Instead, it makes a joint decision by summing the outputs of all experts for a comprehensive judgment.
>
> ### W2: Discussion on the introduction an additional adapter specifically for real facial data.
>
>    We address the practical concern that real-world data can have a wide distribution and include novel scenarios in Section 4.3 ("Effect of Real-LoRA and Fake-LoRAs") and Table 3. Our findings indicate that **expanding a shared Real-LoRA into a sequence of Real-LoRAs, with each module corresponding to the real faces of a specific dataset, yields inferior performance compared to using a single Real-LoRA.** This suggests that real faces exhibit high similarity across datasets, and maintaining a unified distribution for them is more effective than modeling the real faces of each dataset separately.
>
> ### W3: Analysis of the routing distribution.
>    Our MoE framework does not employ an explicit routing strategy. Instead, **it utilizes a form of soft routing constrained by an orthogonality loss, where the final output is generated by directly summing the outputs of the different experts.**
>
>    Due to the time constraints of the rebuttal period, we will add a detailed analysis to the main  paper. This analysis will investigate the behavior of the experts within our proposed framework by examining the output of each expert for different types of attacks and analyzing the orthogonality among them.
>
> ### W4: The reason why the proposed method outperform previous approaches even on a first dataset.
>
>    Our use of a MoE architecture to fine-tune the Feed-Forward Networks (FFN) necessitates the adoption of a Vision Transformer (ViT) framework. During the learning phase on the first dataset (FF++), the strong fitting capability of ViT enables the method to achieve an accuracy of 98.41%. However, models with strong fitting capabilities are prone to overfitting in subsequent tasks and consequently suffer from more severe catastrophic forgetting.
>
>    In continual learning, the forgetting rate is the most important evaluation metric and is relatively less dependent on the backbone architecture. As shown in **Section 4.3, titled "Effect of Lort and Lllb," and in Table 2**, the first row presents the results of using only the baseline. The results indicate that the baseline can achieve a comparable performance (98.18%) on the first task. However, in subsequent tasks, its average accuracy drops sharply, and the forgetting rate increases rapidly, ultimately reaching an Average Accuracy (AA) of only 78.82% and an Average Forgetting (AF) of 23.93%, which is significantly lower than our method's 89.92% AA and 4.03% AF. This finding further isolates the influence of the backbone and demonstrates the effectiveness of our proposed method.
>
> ### W5: The novelty of our framework.
>
>  Although our framework utilizes LoRA as its experts, our core innovations are not LoRA itself, but rather:
>
>    1. Our method for managing the LoRA subspace and the gradient space. This approach prevents interference with the subspaces established for previous tasks throughout the entire training process.
>    2. A label-guided localized balancing strategy. We leverage the key insight that authentic faces exhibit high similarity across datasets, whereas forged faces have significant distributional gaps arising from different manipulation techniques. Our strategy uses this prior to softly assign a shared "Real-LoRA" to model the common features of authentic faces, and a sequence of orthogonal LoRAs to model different forgery types independently and without interference.
>
> ### Limitations.
> We provide a detailed discussion of our framework's limitations in **Appendix A.**

---

> > ### Comment · Reviewer_soP2 · 2025-08-04
> >
> > Thank you for your thorough explanation, which addressed many of my concerns. However, I still have one remaining question:
> >
> > It is mentioned that a single LoRA can model the forged data from a mixed-type dataset and still achieve state-of-the-art results. What is the underlying intuition for this? Additionally, if there are two datasets, A and B, both containing the same deepfake method, how would this affect the output?

---

> > > ### Author Response · Authors · 2025-08-04
> > >
> > > Dear Reviewer soP2,
> > >
> > > We are pleased that our rebuttal has addressed many of your concerns, and we sincerely thank you for your meaningful and insightful comments and interaction.
> > >
> > > In practice, our method establishes a **one-to-one correspondence between Fake-LoRAs and tasks**, where the forgery samples of each task are modeled in a separate LoRA subspace. The operational principle of Fake-LoRAs is as follows: during training, each LoRA, constrained by orthogonality, models both **the common knowledge within the current task's forgery samples** and **the knowledge that is complementary to all previously trained LoRAs**. Our responses to your questions are as follows:
> > >
> > > **(Q1) Mixed-type dataset:** The underlying intuition for applying a single LoRA to model a mixed-type dataset is that this LoRA captures the common knowledge among the various forgery types within that dataset. As long as a LoRA module has sufficient capacity, it can fine-tune the model for multiple forgery types within the same subspace.
> > >
> > > **(Q2) Overlapping Forgery Types:** Each LoRA subspace is orthogonal to all previously established ones. **Therefore, each LoRA, during its training, models knowledge that is complementary to all preceding LoRAs.** In practice, it is not guaranteed that each LoRA will ideally correspond to a distinct forgery type. When two datasets, A and B, contain the same forgery type, T, the new LoRA (for the task with dataset B) learns the knowledge from dataset B that is complementary to what was learned from A. This includes complementary knowledge from the overlapping type T as well as knowledge from any new forgery types present in B. If the data samples for type T are identical in both datasets, the new LoRA receives almost no gradient from the already-learned samples of T.
> > >
> > > **We verify this with a simple experiment under an extreme condition to investigate the effect on the output for an overlapping forgery type.** We use an identical dataset that contains only one forgery type, Deepfakes, to construct two sequential tasks: [Deepfakes, Deepfakes]. When the first LoRA is trained on the first task, the model converges successfully. However, when we deploy a second LoRA for the second task using the identical data, we observe a different behavior. The orthogonality constraints, combined with the zero-initialization of the dimensionality expansion matrix A, cause the training to begin with a very low loss. The gradient is extremely small and shows almost no descent. Consequently, the new LoRA consistently maintains a minimal output. This demonstrates that a new LoRA primarily models knowledge complementary to preceding LoRAs. When the data is identical, the new LoRA cannot acquire sufficient gradients for learning and maintains a low output.

---

> > > > ### Comment · Reviewer_soP2 · 2025-08-05
> > > >
> > > > Thank you for your clear explanation. In light of your comments, I have decided to revise my rating to borderline accept.

---

### Official Review · Reviewer_gTcD · 2025-07-06

**Clarity:** 3
**Significance:** 3
**Originality:** 2
**Rating:** 4
**Confidence:** 3

**Summary:**

This submission tries to integrate continual learning into the detection of generated faces. It uses a mixture of expert model architecture with some modifications, i.e., a real LoRA and a set of orthogonal LoRAs for fake faces. Besides, orthogonal gradient loss was formulated to alleviate inference. Experimental results showed the proposed method had effecive detection scores and low forgetting rates.

**Questions:**

Please see Weakness.

**Ethical Concerns:**

["NO or VERY MINOR ethics concerns only"]

**Final Justification:**

The authors included the experimental comparisons with the suggested one, which helped resolve my concerns in part. Thus, I slightly raised my rating accordingly.

**Limitations:**

NA.

**Paper Formatting Concerns:**

NA.

**Quality:**

2

**Strengths And Weaknesses:**

Strength:

1) Continual learning was integrated into the synthetic face detection task, and was showed effective to alleviate forgetting encountered with unseen types of fake faces.

2) LoRAs were split into real LoRA and a set of fake LoRAs, where fake ones were designed to be orthogonal to each other to capture incremental fake face information. Optimization strategies were designed to improve its training.

Weakness:

1) The proposed method should be compared with more state-of-the-art deepfake video detectors, e.g. [1].

2) MoE has been used in existing deepfake image detection, e.g., [2,3], and the orthogonal gradient optimization has been used in [4] for deepfake audio detection. These works should be discussed regarding the major differences.


References.
[1] Generalizing Deepfake Video Detection with Plug-and-Play: Video-Level Blending and Spatiotemporal Adapter Tuning, https://arxiv.org/abs/2408.17065.

[2] Liu, Zihan, Hanyi Wang, Yaoyu Kang, and Shilin Wang. "Mixture of low-rank experts for transferable ai-generated image detection." arXiv preprint arXiv:2404.04883 (2024).

[3] Cao, Huangsen, Yongwei Wang, Yinfeng Liu, Sixian Zheng, Kangtao Lv, Zhimeng Zhang, Bo Zhang, Xin Ding, and Fei Wu. "HyperDet: Generalizable Detection of Synthesized Images by Generating and Merging A Mixture of Hyper LoRAs." arXiv preprint arXiv:2410.06044 (2024).

[4] Chen, Yujie, Jiangyan Yi, Cunhang Fan, Jianhua Tao, Yong Ren, Siding Zeng, Chu Yuan Zhang et al. "Region-based optimization in continual learning for audio deepfake detection." In Proceedings of the AAAI Conference on Artificial Intelligence, vol. 39, no. 22, pp. 23651-23659. 2025.

---

> ### Author Rebuttal · Authors · 2025-07-30
>
> **Thank you for your thoughtful and constructive review. Our responses to your questions are as follows:**
> ### W1: Comparison with more state-of-the-art deepfake video detectors, e.g. [1].
>
> We provide a comparison following the same experimental settings as DFIL, using the AUC as the evaluation metric. The results are presented in the table below:
>
> | Dataset  | DFDC-P | DFD   | CDF2  |
> | :------- | ------ | ----- | ----- |
> | VB-StA   | 90.9   | 96.5  | 94.7  |
> | **Ours** | 96.21  | 98.71 | 94.93 |
>
> The results demonstrate that our method outperforms the VB-StA[1] method. We will incorporate an additional experiment in the main paper to provide a more detailed comparison. It is crucial to highlight that the method you have requested for comparison is a non-continual learning approach, which is trained on a fixed dataset and evaluated on other datasets. In contrast, our method follows a continual learning paradigm, where the model is sequentially trained and tested on a stream of tasks, emphasizing its ability to mitigate catastrophic forgetting. The training methodologies and objectives of these two approaches are fundamentally different. To ensure a fair comparison, we strictly adhere to the unified testing protocol established in the published DFIL paper, which is designed to evaluate both continual and non-continual learning methods under the same benchmark.
>
> ### W2: Major differences between our method and prior works that utilize Mixture-of-Experts (MoE) and orthogonal optimization.
>
> First, concerning the MoE-based methods in [2,3], the fundamental differences from our approach are as follows:
>
> 1. **These methods employ static, pre-defined Mixture of Experts (MoE) architectures that are unsuitable for continual learning settings.** For instance, [2] pre-defines a set of LoRAs as experts to extract and fuse diverse image features, while [3] uses a similar pre-defined LoRA structure to enhance model generalization. In these frameworks, the LoRAs are trained synchronously and cannot be specialized for individual forgery types, leading to poor scalability when encountering continually emerging manipulation techniques. Consequently, performance degrades significantly when a forgery type appears that is beyond the scope of the pre-defined experts.
>
>    In contrast, our proposed MoE architecture is an extensible framework designed for continual learning. When a new forgery type is introduced, our framework dynamically adds a new LoRA to adapt to it. To preserve previously acquired knowledge, we enforce an orthogonality constraint between the new and existing LoRAs.
>
> 2. **These methods lack designs and priors specifically tailored for continual face forgery detection.** Our approach, however, leverages a key characteristic of this task: real faces exhibit high similarity due to consistent acquisition methods, whereas forged faces show significant variance across different manipulation techniques. Based on this prior, we devise a label-guided LoRA assignment strategy. A shared LoRA is used to model the common features of real faces, while a unique, orthogonal LoRA is assigned to model each specific type of forgery. As demonstrated in Section 4.3, "Ablation Study: Effect of Real-LoRA and Fake-LoRAs," and Table 3, the configuration with Real-LoRA=0 and Fake-LoRAs=4 represents the baseline performance when applying LoRA and MoE directly, without our Real-LoRA and label-guided assignment strategy. The results show that with our innovations, the average task forgetting rate is reduced by 2.39%. We will include the two aforementioned methods in our performance comparison if they become open-source in the future.
>
> To the best of our knowledge, our work is the first to apply an MoE framework to continual face forgery detection.
>
> Second, regarding the orthogonal gradient optimization method in [4], the key distinction is:
>
> 1. **The gradient optimization in [4] is applied to the entire model and the full gradient space. However,our method confines the optimization to the LoRA subspaces.** We aim to use an orthogonal gradient constraint to prevent the new LoRA from interfering with the subspaces of previous tasks during training. Therefore, instead of simply applying an orthogonal loss to the entire gradient space, we introduce an SVD-based decomposition to align the dimensions of the gradient space with the LoRA subspace. This technique preserves the principal components of the gradient and orthogonalizes them with respect to the subspaces of previous tasks.
>
> Similarly, we will provide a detailed discussion in the Related Works section to differentiate our proposed method from these three approaches.

---

> > ### Comment · Reviewer_gTcD · 2025-08-05
> >
> > I thank the authors for preparing the rebuttal, which partially resolved my concerns. I have checked the responses and comments from other reviewers, I have raised my score slightly. Regarding the responses, I don't agree with the claim that "real faces exhibit high similarity due to consistent acquisition methods". The authors used a t-SNE visualization as evidence; however, it's not safe at all to arrive at this conclusion with this mere observation.  Besides, intuitively, the similarity should also depend on how the face images were manipulated and the portion of the manipulation. In addition, though I agree that introducing continual learning might be a contribution, given that this technique could be integrated to any other learning tasks, the novelty might be compromised.

---

> > > ### Author Response · Authors · 2025-08-07
> > >
> > > Dear Reviewer gTcD,
> > >
> > > We are very pleased to know that our rebuttal has addressed some of your concerns, and we sincerely appreciate your willingness to raise the score. We also thank you for the constructive concerns you raised in your response. The discussions with the reviewers have been invaluable for further improving our paper. We have since made new experimental progress in validating our motivation and novelty, which we would like to share with you in hopes of resolving your remaining questions.
> > >
> > > Regarding our motivation, we were initially inspired by a t-SNE experiment, which suggested that abundant and unbiased real faces exhibit a more compact distribution across datasets compared to forged faces created by various methods. **This inspirational hypothesis has now been further validated by a new form of experiment.** As we reached a consensus with Reviewer soP2 in our discussion, within a sequence of orthogonal LoRA modules, **a new LoRA models knowledge that is complementary to all preceding LoRAs** under the constraints of orthogonal gradients and subspaces. When a new orthogonal LoRA is configured to model a repeated domain, ***this domain will have an identical or highly similar input space, $H_{t} \in \mathbf R^{d_{I} \times n}$. From Equation 7, it can be deduced that the resulting $(V_t)\_r$ will also be highly similar. Consequently, in the orthogonal gradient calculation (Equation 9), the term $\\sum \\Vert (V\_{t})\_{r} B\_{i} \\Vert^{2}$ will yield a near-zero gradient because its components are already approximately orthogonal.*** This leads to a small orthogonal gradient loss that is difficult to minimize, making it challenging for the new LoRA to learn from the repeated domain. A more detailed description of this phenomenon can be found in our discussion with Reviewer soP2.
> > >
> > > Based on this, we designed a new experiment composed of two separate orthogonal LoRA sequences: **one for modeling real faces and another for modeling forged faces.** These sequences learn from the task series [FF++, DFDCP, DFD, CDF2], and we recorded the **approximate values of the orthogonal gradient loss for both sequences** to investigate the cross-dataset similarity of the input spaces. The experimental results are presented in the table below:
> > >
> > > | Orthogonal Gradients Loss                  | FF++           | DFDCP          | DFD            | CDF2           |
> > > | ------------------------------------------ | -------------- | -------------- | -------------- | -------------- |
> > > | **Orthogonal LoRA Sequence for Real Face** | 1.413 to 0.016 | 0.187 to 0.025 | 0.279 to 0.005 | 0.097 to 0.039 |
> > > | **Orthogonal LoRA Sequence for Fake Face** | 1.364 to 0.010 | 1.807 to 0.002 | 1.966 to 0.022 | 1.236 to 0.045 |
> > >
> > > We observed that, starting from the second task, **the orthogonal gradient induced by the real face data for a new LoRA was approximately $\frac{1}{10}$ of that induced by the forged face data.** We also observed a slower decrease in this loss, indicating that it is more difficult for the new orthogonal LoRA to learn from the real faces in subsequent tasks. The results suggest that **real faces from different datasets share a similar input space $H_{t} \in \mathbf R^{d_{I} \times n}$**, leading to an orthogonal gradient loss that is an order of magnitude smaller. Moreover, the preceding LoRA modules introduce additional computational overhead compared to a single Real-LoRA. **This experiment mutually validates our initial t-SNE results, demonstrating that using a single, shared Real-LoRA to model real faces is both reasonable and necessary.** Due to time constraints, we will incorporate a detailed analysis and quantitative metrics of this experiment into the final version of the paper.

---

> > > > ### Author Response · Authors · 2025-08-07
> > > >
> > > > The analysis above demonstrates the unique properties of face forgery detection: **real faces possess substantial common knowledge across datasets, whereas forged faces exhibit complementary knowledge.** Compared to methods like DFIL, which can be directly applied to binary classification in a continual learning setting, **our approach is designed specifically around these unique data distribution properties of the forgery detection task, making it both innovative and highly task-specific.** Here, we will further elaborate on the novelty of our work. We illustrate this with an experiment that we began during the rebuttal phase but were unable to include previously due to time limits. This experiment shows that **general-purpose continual learning methods using MoE and LoRA do not perform well on the continual face forgery detection task, whereas our proposed strategy of using face detection distribution characteristics to guide different LoRA modules can significantly improve performance.** Specifically, we applied the general-purpose continual learning methods, InfLoRA, MoECL and O-LoRA (which was included in the comparisons in our paper), to the continual face forgery detection task and compared them with our DevFD, as shown in the table below:
> > > >
> > > > | Datasets        | FF++      | DFDCP     | DFDCP    | DFD       | DFD      | CDF2      | CDF2     |
> > > > | --------------- | --------- | --------- | -------- | --------- | -------- | --------- | -------- |
> > > > | Metrics         | AA        | AA        | AF       | AA        | AF       | AA        | AF       |
> > > > | MoECL           | 98.13     | 90.71     | 6.54     | 89.04     | 9.13     | 83.74     | 12.67    |
> > > > | O-LoRA          | 97.30     | 91.08     | 4.53     | 89.71     | 9.01     | 87.66     | 10.06    |
> > > > | InfLoRA         | 97.86     | 90.19     | 5.33     | 90.37     | 5.27     | 85.44     | 7.05     |
> > > > | **Ours(DevFD)** | **98.41** | **93.48** | **1.35** | **93.00** | **3.45** | **90.58** | **3.72** |
> > > >
> > > > We found that, lacking designs tailored to the specific characteristics of the continual forgery detection task, these **general-purpose SOTA methods fail to achieve satisfactory results when directly applied**. **Our method, designed specifically for the distributional properties of real and forged faces, achieves performance far superior to these general SOTA continual learning methods.** We believe this strongly demonstrates the **novelty** and **task-specific effectiveness** of our approach for continual face forgery detection. We will also incorporate this experiment into the main paper.
> > > >
> > > > Thank you once again for your feedback on our method and rebuttal, and for raising your score.
> > > >
> > > > Sincerely,
> > > >
> > > > The Authors

---

### Note · Authors · 2025-08-12

Dear AC and Reviewers,
We thank you all for your valuable suggestions to improve this work. We would like to further clarify the **motivation** and **novelty** of the proposed asymmetric MoE structure:

- **Motivation:** Our work is motivated by a key observation:  the distribution of real faces is relatively similar across different datasets, while forged faces exhibit significant differences due to varying creation methods. We substantiate this observation with the following evidence:
  1. **Data Acquisition:** When compiling deepfake datasets, real faces are typically gathered without introducing specific biases (e.g., race, gender), leading to a relatively stable distribution. In contrast, forged faces are generated by a wide range of methods, resulting in more varied distributions.
  2. **Visualization:** The t-SNE visualization in Fig. 1 visually corroborates our observation.
  3. **Input Space Analysis:** We conduct a new experiment (as discussed with Reviewers gTcD and 1J7u) that further validates this observation. The results demonstrate that the input space for real faces exhibits greater similarity across datasets compared to that of forged faces.
  4. **Ablation Study:** Results in Table 3 demonstrate that modeling real faces with a sequence of orthogonal LoRA modules does not yield performance gains over a shared LoRA.

- **Novelty & Task-specific Innovations:** Our framework is not a mere assemblage of existing MoE structures or a straightforward application of generic methods, but is **specifically designed for deepfake detection:**
  1. **Difference in the Key Idea:**  Generic CL methods, when applied directly to deepfake detection, learn from both real and fake data in a unified manner. However, our key insight is that real and fake data exhibit distinct properties. Therefore, for each new task, our method employs distinct sub-networks (i.e., experts) to handle real and fake data separately. This disentangled learning enables our sub-networks to specialize in identifying new forgery artifacts.
  2. **Demonstrated Performance:** Our method significantly outperforms SOTA generic CL methods (e.g., MoECL, O-LoRA) in new experiments (discussed with Reviewers gTcD and 1J7u). This demonstrates the practical benefits of our task-specific design.
  3. **In Response to Task-specific Issues:** While our method is tailored for deepfake detection, its potential applicability to other tasks should be recognized as a strength rather than a weakness.

---

### Decision · Program_Chairs · 2025-09-17

**Decision:**

Accept (poster)

**Comment:**

This submission integrates continual learning into the detection of fake faces, using an MoE architecture with one Real-LoRA for real faces and a set of orthogonal Fake-LoRAs for fake faces. Orthogonal gradient loss was formulated to alleviate inference. Experimental results showed the proposed method has higher detection scores and lower forgetting rates.

The main strengthens of the work include: the motivation of the paper is clear, and the proposed solution is simple and effective; integrating orthogonal gradients into the orthogonal loss is a sound technical contribution; the paper is well-structured and easy to follow; the dataset-incremental and manipulation-incremental results are promising; and the ablation studies well demonstrate the effectiveness of the designed components.

The main weaknesses lie in the moderate novelty (LoRA with MoE for CL is not brand new), strong assumption that real faces are stable and can be modeled by one single Real-LoRA. The AC notices that not all the concerns were completely solved. However, compared to the many strengths of the work, the weaknesses can be ignored for the time being, thus deserving public to the community.